# Conditional deletion of neurexins dysregulates neurotransmission from dopamine neurons

Charles Ducrot[1,2,3], Gregory de Carvalho[4], Benoît Delignat-Lavaud[1,2,3], Constantin VL Delmas[5], Priyabrata Halder[1,2,3], Nicolas Giguère[1,2,3], Consiglia Pacelli[6], Sriparna Mukherjee[1,2,3], Marie-Josée Bourque[1,2,3], Martin Parent[5], Lulu Y Chen[4]*, Louis-Eric Trudeau[1,2,3]*

[1]Department of Pharmacology and Physiology, Faculty of Medicine, Université de Montréal, Montréal, Canada; [2]Department of Neurosciences, Faculty of Medicine, Université de Montréal, Montréal, Canada; [3]Neural Signaling and Circuitry Research Group (SNC), Montréal, Canada; [4]Department of Anatomy and Neurobiology, School of Medicine, University of California, Irvine, Irvine, United States; [5]CERVO Brain Research Centre, Department of Psychiatry and Neurosciences, Faculty of Medicine, Université Laval, Quebec, Canada; [6]Department of Clinical and Experimental Medicine, University of Foggia, Foggia, Italy

*For correspondence:
chenly@uci.edu (LYC);
louis-eric.trudeau@umontreal.ca
(L-EricT)

Competing interest: The authors declare that no competing interests exist.

**Abstract** Midbrain dopamine (DA) neurons are key regulators of basal ganglia functions. The axonal domain of these neurons is highly complex, with a large subset of non-synaptic release sites and a smaller subset of synaptic terminals from which in addition to DA, glutamate or GABA are also released. The molecular mechanisms regulating the connectivity of DA neurons and their neurochemical identity are unknown. An emerging literature suggests that neuroligins, trans-synaptic cell adhesion molecules, regulate both DA neuron connectivity and neurotransmission. However, the contribution of their major interaction partners, neurexins (Nrxns), is unexplored. Here, we tested the hypothesis that Nrxns regulate DA neuron neurotransmission. Mice with conditional deletion of all Nrxns in DA neurons (DAT::NrxnsKO) exhibited normal basic motor functions. However, they showed an impaired locomotor response to the psychostimulant amphetamine. In line with an alteration in DA neurotransmission, decreased levels of the membrane DA transporter (DAT) and increased levels of the vesicular monoamine transporter (VMAT2) were detected in the striatum of DAT::NrxnsKO mice, along with reduced activity-dependent DA release. Strikingly, electrophysiological recordings revealed an increase of GABA co-release from DA neuron axons in the striatum of these mice. Together, these findings suggest that Nrxns act as regulators of the functional connectivity of DA neurons.

## Editor's evaluation

In this study, the authors selectively delete the three main genes encoding neurexins from dopamine neurons in mice. The authors find that while dopamine axonal architecture and synaptic ultrastructure are generally unaffected by loss of neurexins, there are changes in dopamine reuptake, amphetamine-induced locomotion, and GABA co-release, and notably, these changes are region specific, with most of the effects observed in the ventral striatum. The results are solid, and these findings are valuable and useful, providing new information regarding the potential roles of neurexins in regulating dopamine neuron output.

**eLife digest** The human brain contains billions of nerve cells, known as neurons, which receive input from the outside world and process this information in the brain. Neurons communicate with each other by releasing chemical messengers from specialized structures, called axon terminals, some of which form junctions known as synapses. These messengers then generate signals in the target neurons.

Based on the type of chemical they release, neurons can be classified into different types. For example, neurons releasing dopamine are considered to act as key regulators of learning, movements and motivation. Such neurons establish very large numbers of axon terminals, but very few of them form synapses.

Specific sets of proteins, including neurexins and neuroligins, are thought to help regulate the activity of the connexions between these neurons. Previous research has shown that when neuroligins were removed from the neurons of worms or mice, it affected the ability of the animals to move. So far, the role of neurexins in managing the connectivity of regulatory neurons, such as those releasing dopamine, has received much less attention.

To bridge this knowledge gap, Ducrot et al. explored how removing neurexins from dopamine neurons in mice affected their behaviour. The experiments revealed that eliminating neurexins did not affect their motor skills on a rotating rod, but it did reduce their movements in response to the psychostimulant amphetamine, a molecule known to enhance dopamine-associated behaviours. The cellular structure of dopamine neurons lacking neurexins was the same as in neurons containing this protein. But dopamine neurons without neurexins were slower to recycle dopamine, and they released a higher amount of the inhibitory messenger GABA. This suggests that neurexin acts as an important suppressor of GABA secretion to help regulate the signals released by dopamine neurons.

These findings set the stage for further research into the role of neurexins in regulating dopamine and other populations of neurons in conditions such as Parkinson's disease, where movement and coordination are affected.

## Introduction

Dopamine (DA) neurons from the ventral tegmental area (VTA) and substantia nigra pars compacta (SNc) project densely to the ventral striatum (vSTR) and to the dorsal striatum (dSTR), respectively (*Descarries et al., 1980*; *Matsuda et al., 2009*) and are critical regulators of basal ganglia functions, motivation, and cognition (*Schultz, 2007*; *Surmeier et al., 2014*). The connectivity of the DA system is predominantly non-synaptic (*Descarries et al., 2008*; *Ducrot et al., 2021*; *Wildenberg et al., 2021*), with a majority of DA-releasing terminals not located in close apposition to a postsynaptic domain (*Caillé et al., 1996*; *Descarries et al., 2008*; *Descarries et al., 1996*; *Descarries and Mechawar, 2000*; *Ducrot et al., 2021*). A smaller synaptic subset of DA neuron terminals has the ability to co-release glutamate or GABA (*Dal Bo et al., 2004*; *Mendez et al., 2008*; *Stuber et al., 2010*; *Sulzer et al., 1998*; *Tritsch et al., 2016*; *Tritsch et al., 2012*).

Despite the functional importance of DA in the brain, little is presently known about the molecular mechanisms underlying the formation and regulation of the complex axonal arbor and release sites established by DA neurons. Only a limited number of studies have until now explored the molecular mechanisms underlying DA release (*Banerjee et al., 2022*; *Banerjee et al., 2020*; *Delignat-Lavaud et al., 2021*; *Ducrot et al., 2021*; *Liu et al., 2018*; *Mendez et al., 2011*; *Robinson et al., 2019*, *Delignat-Lavaud et al., 2023*). Interestingly, a growing body of work suggests that the trans-synaptic cell adhesion neuroligins (NLs) directly or indirectly regulate the connectivity of DA neurons. Impaired DA-mediated motor behaviors were reported in *Caenorhabditis elegans* lacking NL-1 (*Izquierdo et al., 2013*; *Rodríguez-Ramos et al., 2017*). Downregulation of NL-2 in striatal neurons was also suggested to reduce the frequency of contacts between DA neuron axons and the dendrites of striatal neurons (*Uchigashima et al., 2016*). Mutations in NL-3 in mice lead to impaired synaptic inhibition onto striatal D1 DA receptor expressing neurons (*Rothwell et al., 2014*). Finally, both NL-1 and NL-2 are permissive for the formation of synapse-like contacts by DA neuron axons (*Ducrot et al., 2021*; *Uchigashima et al., 2016*).

Although NLs mediate some of their cellular and synaptic effects by interacting with neurexins (Nrxns) (*Chen et al., 2017*; *Zhang et al., 2015*), the role of this family of presynaptic cell adhesion molecules in DA neurons is presently unexplored. Nrxns were initially identified as α-latrotoxin receptors (*Ushkaryov et al., 1992*). In mammals, Nrxns are expressed in two principal forms: longer α-Nrxns isoforms and shorter β-Nrxns isoforms (*Tabuchi and Südhof, 2002*). The Nrxn proteins on axon terminals interact with postsynaptic NL proteins and have been shown to regulate synapse formation and function (*Graf et al., 2004*; *Ichtchenko et al., 1995*; *Ko et al., 2009*). NLs only bind to Nrxns, whereas Nrxns have large numbers of splice variants with differential binding affinities with multiple postsynaptic partners. Several key studies using a strategy of conditional Nrxns deletion in mice demonstrated that Nrxns regulate neurotransmission through different mechanisms in a cell type-specific manner (*Chen et al., 2017*; *Luo et al., 2021*; *Luo et al., 2020*).

Here, we tested the hypothesis that Nrxns play a key role in regulating the connectivity and functions of DA neurons by deleting all Nrxns in these cells. We crossed DAT-IRES-Cre mice with Nrxn123α/β floxed mice (Nrxn123 triple conditional KO mice [cKO] [*Chen et al., 2017*; *Figure 1—figure supplement 1*]). We found that these mice show impaired amphetamine-induced locomotion but unaltered synapse ultrastructure. DA signaling was impacted after the loss of Nrxns, as revealed by slower DA reuptake, decreased density of DA transporter (DAT), increased density of vesicular monoamine transporter (VMAT2) and reduced activity-dependent DA release. Finally, electrophysiological recordings showed an increase in GABA release from DA terminals in the vSTR but not dSTR in KO mice, suggesting a region-specific regulatory role of Nrxns on GABA co-transmission in DA neurons.

## Results

### Deletion of Nrxns reduces amphetamine-induced locomotion without affecting basal motor activity or coordination

Previous work provided evidence for the presence of *Nrxn* mRNA in mesencephalic DA neurons (*Uchigashima et al., 2019*; *Uchigashima et al., 2016*). However, no study has previously compared the levels of expression of each Nrxn in this neuronal population. Here, we examined this by measuring mRNA purified from postnatal VTA or SNc DA neurons obtained from transgenic mice expressing the green fluorescent protein (GFP) gene under control of the tyrosine hydroxylase (TH) promoter (TH-GFP mice) by using fluorescence activated cell sorting (FACS) and RNASeq. We found that while mRNA for all three forms of Nrxn are found at high levels in both VTA and SNc DA neurons, *Nrxn1* is found at higher levels in VTA neurons, *Nrxn2* is similarly expressed in both populations and *Nrxn3* is found at higher levels in SNc neurons (*Table 1*). Validating the precision of VTA and SNc dissections, we found that mRNA of the transcription factor Sox6 was found at higher levels in SNc neurons, while that of the vesicular glutamate transporter VGLUT2 and of the calcium binding protein calbindin-1 were found at higher levels in VTA neurons, in line with previous work (*Table 1*; *Dal Bo et al., 2004*; *Panman et al., 2014*; *Pereira Luppi et al., 2021*; *Poulin et al., 2014*). Next, with the objective

**Table 1.** All three neurexins (Nrxns) are expressed in dopamine (DA) neurons.
Table showing Nrxn, Sox6, Slc17a6, and Calbn1 mRNA levels determined by RNASeq in fluorescence activated cell sorting (FACS)-purified ventral tegmental area (VTA) and substantia nigra pars compacta (SNc) DA neurons. Results are presented as FKPM (fragments per kilobase of transcript per million fragments) values. Each value is the average of three independent samples. The statistics refer to the difference between SNc and VTA, determined using a t-test.

| mRNA levels | VTA DA neurons | SNc DA neurons | Statistics |
|---|---|---|---|
| *Nrxn1* | 12,280±369 | 10,698±-359 | p<0.005 |
| *Nrxn2* | 13,505±1042 | 14,038±82 | p>0.05 |
| *Nrxn3* | 7854±295 | 12,857±239 | p<0.001 |
| *Sox6* | 195±17 | 2862±63 | p<0.001 |
| *Slc17a6* | 1953±144 | 587±34 | p<0.001 |
| *Calb1* | 8373±-519 | 1791±49 | p<0.001 |

of understanding the canonical function of all Nrxns in DA neurons, we selectively deleted Nrxn 1, 2, and 3 from DA neurons by crossing Nrxn123$^{flox/flox}$ mice with DAT-IRES-Cre mice (DAT::NrxnsKO; *Figure 1—figure supplement 1*) and examined in male mice the global functional impact of this gene deletion by quantifying motor behaviors. DA neurons are key regulators of movement, motivation, and reward-dependent learning and several studies using mouse lines with impaired DA transmission reported deficits in basal or psychostimulant-evoked locomotion and learning on the accelerating rotarod (*Birgner et al., 2010*; *Ogura et al., 2005*; *Zhou and Palmiter, 1995*).

In the first series of experiments, we evaluated motor coordination and learning using the accelerating rotarod task with two different protocols (*Figure 1A* and *Figure 1—figure supplement 2A*). The first protocol evaluated the rate of learning to perform this task over a total of nine sessions in 3 days, with two sessions performed on the first day, three sessions on the second day, and four sessions on the third day, with a speed of rotation accelerating from 4 to 40 rpm over 10 min. The measure of latency to fall did not reveal a significant difference between the genotypes, with all groups showing a comparable increase in performance (*Figure 1B*, two-way repeated measures ANOVA, $F_{(1, 14)}=1.43$, $p=0.25$). An analysis of the slope of the change in performance across the nine sessions similarly did not reveal any difference between the genotypes in the latency to fall (simple linear regression, $F_{(1, 140)}=0.56$, $p=0.45$, results not shown). Similar results were obtained when evaluating the progression of the performance of the mice by comparing the first and last sessions, with mice of both genotypes showing equivalent learning (*Figure 1C*; two-way ANOVA, main effect of training session, $F_{(1, 28)}=21.72$, $p<0.0001$; Sidak's multiple comparisons test, S1 vs S9: WT, $p=0.022$ and KO, $p=0.001$). The speed of rotation at the end of each trial across all nine trials was also unchanged (*Figure 1D*; two-way repeated measures ANOVA, $F_{(1, 14)}=1.44$, $p=0.25$). A separate cohort of mice were tested using a more challenging version of the rotarod (*Figure 1—figure supplement 2A*), with speed of rotation accelerating from 4 to 40 rpm over 2 min. In this cohort, the latency to fall was not significantly different in DAT::NrxnsKO compared to DAT::NrxnsWT, although a tendency for impaired performance was observed (*Figure 1—figure supplement 2B*, two-way ANOVA, repeated measures, $F_{(1, 16)}=4.00$, $p=0.06$). In this task, performance failed to improve over the trials, revealing a limited capacity to improve performance, as shown by comparing performance in the first and last sessions (*Figure 1—figure supplement 2C*; two-way ANOVA, $F_{(1, 32)}=0.037$, $p=0.84$). The speed of rotation at the end of each trial across all nine trials (*Figure 1—figure supplement 2D*) was similar in DAT::NrxnsKO mice compared to the control mice (two-way repeated measures ANOVA, $F_{(1, 16)}=3.50$, $p=0.08$). These results suggest that deletion of Nrxn123 from DA neurons does not lead to major motor coordination and motor learning deficits.

General motor abilities were next evaluated using the pole test and the open field test. In the pole test, no difference was observed between genotypes for the time required for the mice to orient downward (*Figure 1—figure supplement 2E*; unpaired t-test, $p=0.15$) but interestingly the time required to climb down the pole was significantly higher for the DAT::NrxnsKO mice (*Figure 1—figure supplement 2F*; unpaired t-test, $p=0.034$).

Basal locomotion in the open field over a 60 min period was also not different between genotypes (*Figure 1E*; two-way ANOVA, repeated measures, $F_{(1; 18)}=3.77$, $p=0.068$). We next challenged the dopaminergic system of these mice using the psychostimulants cocaine and amphetamine (*Di Chiara and Imperato, 1988*). Although locomotion induced by cocaine (20 mg/kg) was comparable between genotypes (*Figure 1F*, two-way ANOVA, repeated measures, $F_{(1; 16)}=0.64$, $p=0.43$), locomotion induced in response to amphetamine (5 mg/kg) was strongly reduced in DAT::NrxnsKO mice compared to DAT::NrxnsWT mice (*Figure 1G*, two-way ANOVA, repeated measures, $F_{(1; 13)}=6.66$, main effect of genotype, $p=0.023$). The finding of reduced behavioral response to amphetamine suggests that loss of Nrxns in DA neurons leads to some alteration of the functionality of the DA system and some DA-dependent behaviors.

Because altered DA neurotransmission is often associated with changes in states of hedonia, we next examined the performance of the mice in a well-established sucrose preference task (*Figure 1—figure supplement 2G*). On the initial two conditioning days (CD1 and CD2), mice of all genotypes equally licked at both bottles (*Figure 1—figure supplement 2H*). Similarly, during the next three testing days (TD1, -2, and -3), when mice were given a choice between water and sucrose, DAT::NrxnsKO and WT mice both showed a similar marked preference for the sucrose bottle (*Figure 1—figure supplement 2H*; two-way ANOVA, main effect of choice $F_{(3; 20)}=487.0$; Tukey's multiple

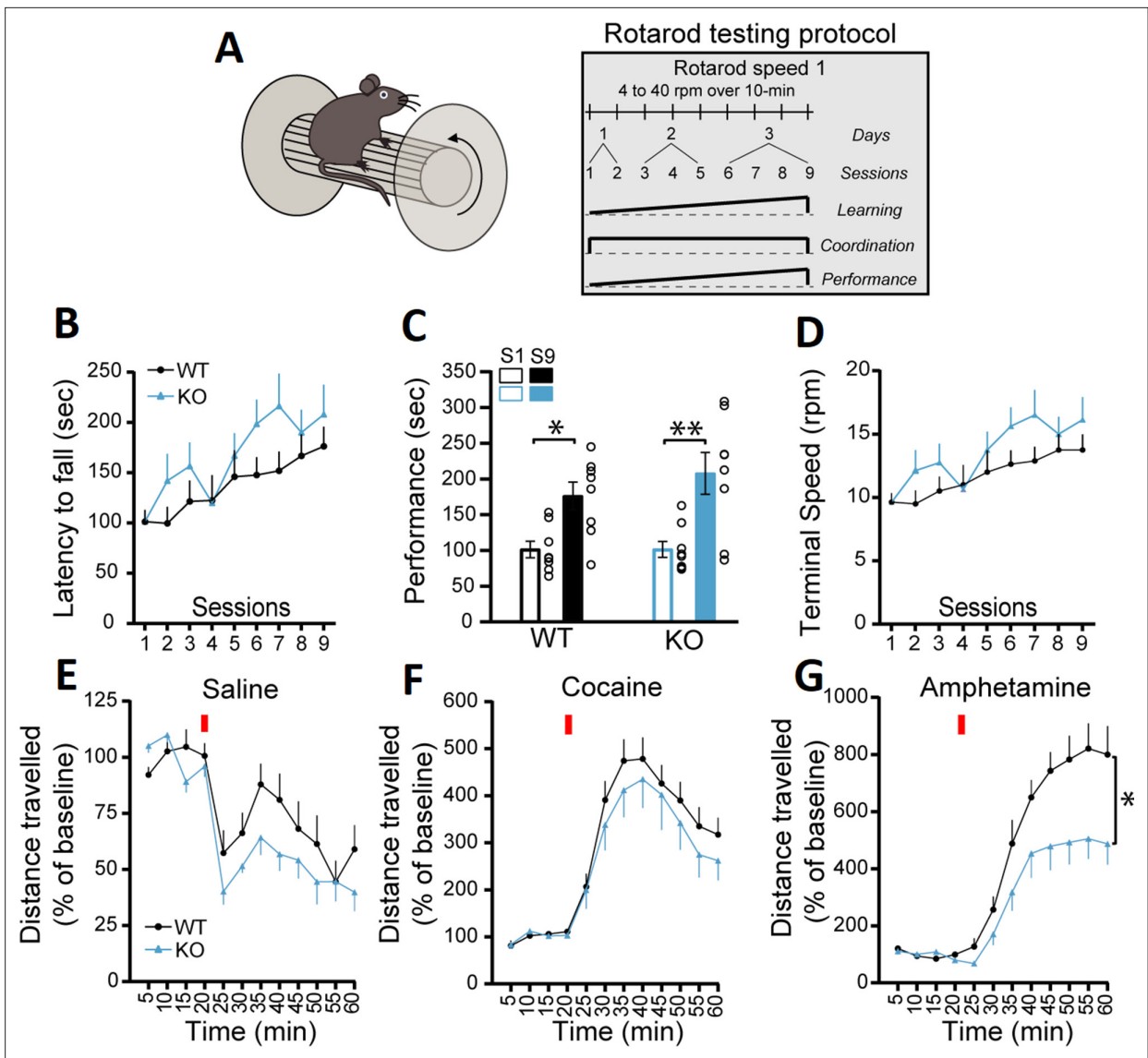

**Figure 1.** DAT::NrxnsKO mice exhibit impaired amphetamine-induced motor activity. (**A**) Schematic representation of a mouse on a rotarod and the diagram of the rotarod testing protocol for the speed 1. (**B**) Performance on the accelerating rotarod during nine sessions over 3 consecutive days. Latency to fall was quantified at rotation speeds from 4 to 40 rpm over 10 min. (**C**) Performance of DAT::NrxnsKO and WT littermate mice on the rotarod was evaluated comparing the last session and the first session for each mouse. The results show a significant improvement in performance irrespective of genotype. (**D**) Quantification of the terminal speed over all the sessions shows no difference between the DAT::NrxnsKO and WT littermate mice. (**E**) Basal horizontal activity in a novel environment before and after a saline injection (10 mL/kg) over a total of 60 min. (**F**) Horizontal activity before and after a cocaine injection (20 mg/kg; 10 mL/kg) over a total of 60 min. (**G**) Horizontal activity before and after an amphetamine injection (5 mg/kg; 10 mL/ kg) over 60 min shows reduced locomotion in the DAT::NrxnsKO compared to the control mice. For rotarod and locomotor activity experiments, 7–10 animals per group were used. For all analyses, the plots represent the mean ± SEM. Statistical analyses were carried out by two-way ANOVAs followed by Tukey's multiple comparison tests or Sidak's multiple comparisons test. The stars in panel D represent the level of significance of the post hoc tests (*p<0.05; **p<0.01).

The online version of this article includes the following source data and figure supplement(s) for figure 1:

**Source data 1.** Contains the primary data for *Figure 1* and *Figure 1—figure supplement 2*.

**Figure supplement 1.** Breeding scheme for generation of DAT::NrxnsKO; DAT::NrxnsWT, and DAT::NrxnsHET.

**Figure supplement 2.** Lack of notable changes in the behavioral performance of DAT::NrxnsKO mice.

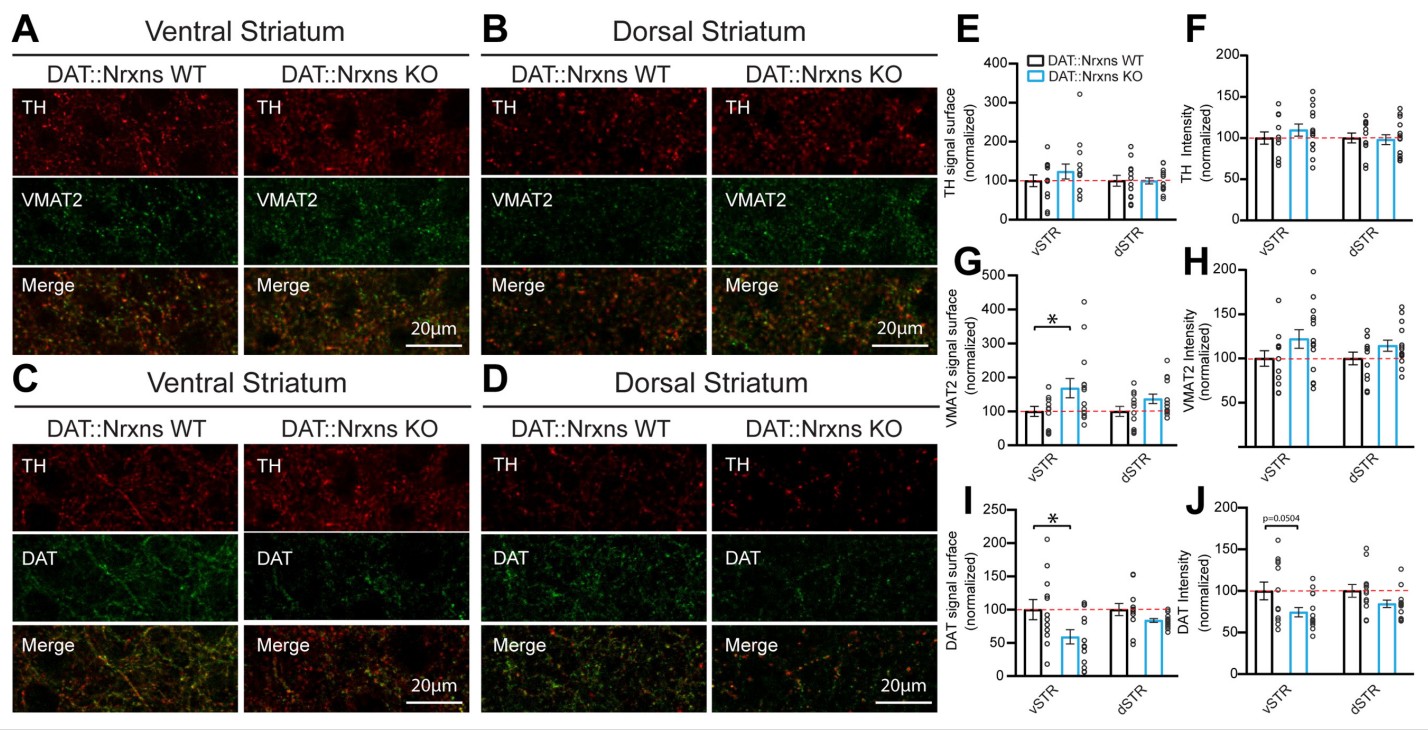

**Figure 2.** Increased vesicular monoamine transporter (VMAT2) but decreased dopamine transporter (DAT) expression in dopamine (DA) axon terminals lacking neurexins (Nrxns). (**A** and **B**) Immunohistochemistry characterization of ventral (**A**) and dorsal (**B**) striatal slices from 8-week-old DAT::NrxnsKO and DAT::NrxnsWT mice (60× confocal) using tyrosine hydroxylase (TH, red) and VMAT2 (green) antibodies. (**C** and **D**) Immunohistochemistry of ventral (**C**) and dorsal (**D**) striatal slices from DAT::NrxnsKO and DAT::NrxnsWT mice using TH (red) and DAT (green) antibodies. (**E–J**) Quantification of signal intensity and signal surface (% of WT) for TH, VMAT2, and DAT in the different striatal regions examined: ventral striatum (vSTR) and dorsal striatum (dSTR) (DAT::NrxnsKO = 14 hemispheres/7 mice; DAT::NrxnsWT = 12 hemispheres/6 mice). TH surface area: vSTR = 123.6 ± 18.99% and dSTR = 99.49 ± 7.73% of control. TH signal intensity: vSTR = 109.6 ± 7.36% and dSTR = 97.96 ± 5.98% of control. VMAT2 surface area: vSTR = 168.3 ± 28.27% and dSTR = 136.7 ± 13.85% of control. VMAT2 signal intensity: vSTR = 122.1 ± 10.48% and dSTR = 114.4 ± 6.25% of control. DAT surface area: vSTR = 59.00 ± 10.71% and dSTR1=83.70 ± 2.70% of control DAT signal intensity: vSTR = 74.37 ± 5.56% and dSTR = 84.42 ± 4.40% of control. Statistical analysis was carried out by unpaired t-test for each substructure. Surface and intensity for each signal were measured in striatal slice from bregma + 0.74 mm, with a total of seven different spots for each hemisphere from six DAT::NrxnsWT mice and seven DAT::NrxnsKO mice. Error bars represent ± SEM (*p<0.05).

The online version of this article includes the following source data for figure 2:

**Source data 1.** Contains the primary data for *Figure 2*.

comparisons test, TD1, -2, -3, water versus sucrose, all genotypes, p<0.0001). These findings suggest that the response of DAT::NrxnsKO mice to natural rewards was unaltered.

## Altered DAT and VMAT2 levels in the vSTR confirm a perturbation of the DA system in Nrxns KO mice

The reduced locomotor response to amphetamine in DAT::NrxnsKO mice suggests the possibility that the structure or the function of DA neurons or their terminals in the striatum are altered in the absence of Nrxns. First, we performed immunohistochemistry to examine the levels of the DA biosynthetic enzyme TH, the VMAT2, and the membrane DAT. The immunopositive surface area of these markers was quantified in a series of three striatal brain sections ranging from bregma +0.74 to bregma –0.82 mm, with a total of seven different regions in each hemisphere distributed to cover the ventral and dorsal sectors of the striatum. We found that TH surface area was unchanged in both the vSTR and dSTR (*Figure 2A–B , and E*). However, the surface of VMAT2 immunoreactivity was significantly increased in the vSTR, but not in the dSTR, of the KO mice (*Figure 2A–B , and G*; vSTR, unpaired t-test, Welch's corrected, p=0.045). In contrast, DAT surface area was significantly decreased in the vSTR, but not in the dSTR, of the KO mice (*Figure 2C and F*; vSTR, unpaired t-test, p=0.034).

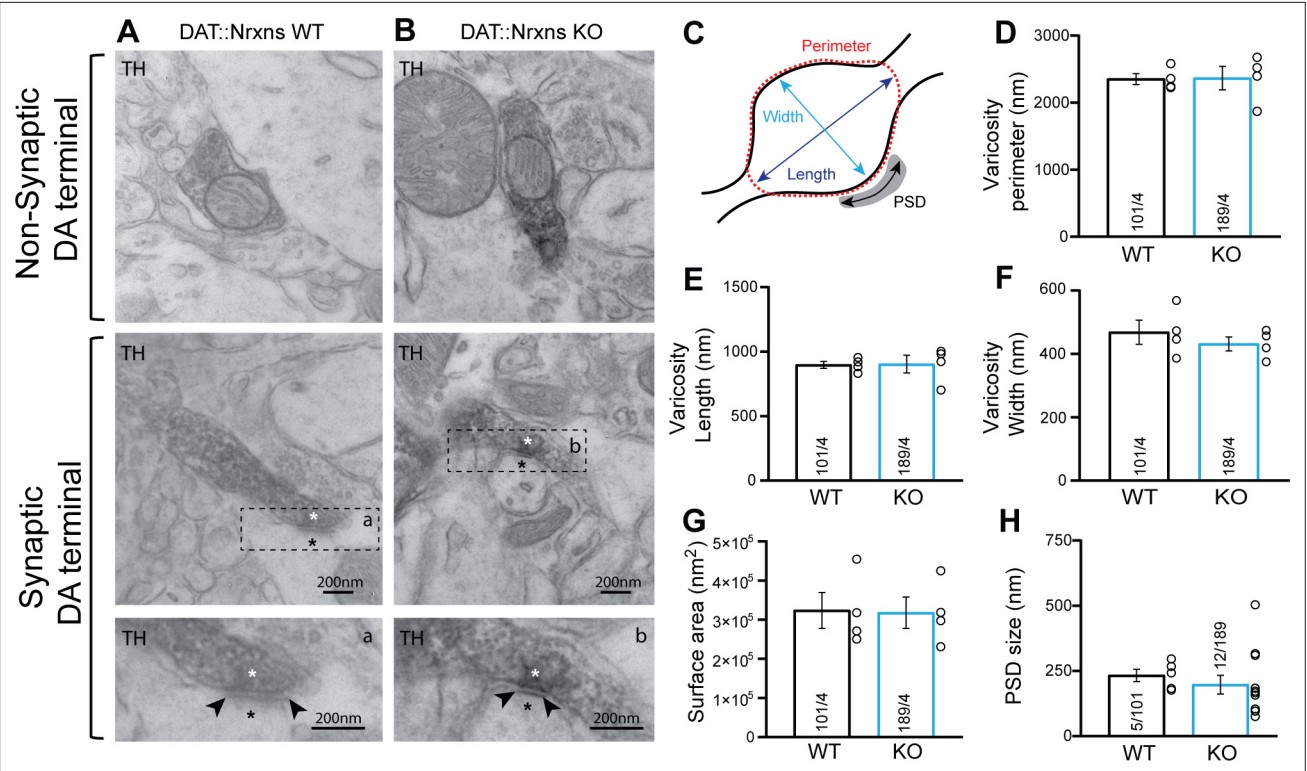

**Figure 3.** Synaptic and non-synaptic ultrastructure of dopamine (DA) terminals is unchanged after the deletion of neurexins (Nrxns) in DA neurons. (**A–B**) Electron micrographs showing DA neuron terminals without any postsynaptic density (PSD) domain (top images) or in apposition to a PSD domain in ventral striatal tissue from DAT::NrxnsWT and KO mice. The lower micrograph represents a magnified view of the regions identified by the doted lines in the middle images. The asterisk identifies a synapse and the black arrowheads delimitate the postsynaptic domain. (**C**) Schematic representation of a dopaminergic varicosity. (**D**) Bar graph representing the perimeter of the DA axonal varicosity from WT and KO mice (2353±81.83 nm and 2366±174.8 nm, respectively). (**E** and **F**) Bar graphs representing the size of the axonal varicosities, quantified as length (**E**) (897.3±38.06 nm and 902.7±38.06 nm, respectively) and width (**F**) (468.7±38.06 nm and 431.5±22.02 nm, respectively). (**G**) Bar graphs showing the surface area of DA neuron varicosities from WT and KO animals (323,537±45,861 $nm^2$ and 317,887±40,227 $nm^2$, respectively). (**H**) Bar graphs representing the PSD domain size from individual synapses (232.8±23.40 nm and 197.1±35.71 nm, respectively, for WT and KO mice). For all analyses, WT = 101 and KO = 189 axonal varicosities from four different mice for each genotype. For all analyses, plots represent the mean ± SEM. Statistical analyses were carried out by unpaired t-tests.

The online version of this article includes the following source data for figure 3:

**Source data 1.** Contains the primary data for *Figure 3*.

In addition to the surface of immunopositive signal, the average intensity was also quantified. TH, VMAT2, and DAT signal intensity of DAT::NrxnsKO mice were unchanged in both vSTR and dSTR (*Figure 2A–D, F, H and J*).

## Nrxn123 ablation does not impair synapse ultrastructure in DA neurons

The changes in DAT and VMAT2 immunoreactivity could represent changes in protein expression or axon terminal density or structure. To gain insight into this, we next examined the integrity of axon terminals and synapses established by DA neurons in the intact brain by transmission electron microscopy (TEM). We focused on terminals in the vSTR, where we observed significant changes in VMAT2 and DAT, which is the most characterized brain region showing DA neuron-mediated glutamate and GABA co-transmission (*Bérubé-Carrière et al., 2012*; *Stuber et al., 2010*). Our results show that, irrespective of the genotype, most axonal varicosities contained synaptic vesicles and mitochondria (*Figure 3A–B*). Furthermore, TH-positive dopaminergic terminals in the vSTR of DAT::NrxnsKO mice were not different compared to DAT::NrxnsWT mice in terms of their overall perimeters (P) (*Figure 3C–D*; unpaired t-test, p=0.94), length (L) (*Figure 3E*, unpaired t-test, p=0.94), width (w) (*Figure 3F*, unpaired t-test, p=0.43), or surface area (*Figure 3G*, unpaired t-test, p=0.92).

In addition, the propensity of these terminals to make contact with a postsynaptic density (PSD) domain was unchanged in DAT::NrxnsKO mice. The synaptic incidence of TH-positive terminals was 6.34% (12 terminals with a PSD domain/189 TH-positive varicosities) for DAT::NrxnsKO mice and 4.95% (5 terminals with a PSD domain/101 TH-positive varicosities) for control mice (data not shown), a low proportion in line with previous work (*Bérubé-Carrière et al., 2012*; *Stuber et al., 2010*). Among these synaptic TH-positive varicosities, the size of the PSD was similar (*Figure 3H*, unpaired t-test, p=0.54). Together, these results show that loss of Nrxns123 does not impair the basic ultrastructure of DA release sites in the vSTR.

## FSCV reveals altered DA release parameters after conditional deletion of all Nrxns in DA neurons

The impaired response to amphetamine suggests a perturbation of extracellular DA dynamics or DA action on target cells. To examine this possibility, we first employed fast-scan cyclic voltammetry (FSCV) in acute brain slices of the vSTR and dSTR to measure electrically evoked DA overflow (DAo), the identify of which was confirmed by the shape of cyclic voltamograms (*Figure 4—figure supplement 1A–D*). In the first series of experiments performed in normal extracellular saline, we found no difference in peak DAo between the DAT::NrxnsWT and DAT::NrxnsKO mice (*Figure 4A–B and E–F*).

However, an examination of the kinetics of DAo, often used to identify changes in DA release efficiency and reuptake (*Yorgason et al., 2011*), revealed that the tau value of DA recovery in the vSTR was significantly higher in DAT::NrxnsKO compared to DAT::NrxnsWT mice (*Figure 4A and C*, unpaired t-test, p=0.008). We also observed a similar increase in tau for the rate of DA recovery in the dSTR (*Figure 4E and G*; unpaired t-test, p=0.019). Quantification of the rise time of evoked DAo in the vSTR and dSTR revealed no significant genotype difference (*Figure 4—figure supplement 2A–B*).

Short-term plasticity of electrically evoked DA release in the striatum was examined using a paired-pulse stimulation paradigm. DAo in acute brain slices typically shows a large paired-pulse depression, more extensively so in the dSTR compared to the vSTR (*Condon et al., 2019*; *Sanchez et al., 2014*; *Zhang and Sulzer, 2004*). Interestingly, the level of paired-pulse depression was significantly enhanced in DAT::NrxnsKO mice compared to WT in the vSTR (*Figure 4D*; unpaired t-test, p=0.04). However, in the dSTR, paired-pulse depression was similar in DAT::NrxnsKO mice compared to WT mice (*Figure 4H*; unpaired t-test, p=0.62). Plasticity of DA release was also examined by measuring the inhibition of DAo induced by the GABA$_B$ agonist baclofen (10 μM) (*Lopes et al., 2019*). A recent study demonstrated a role for Nrxns in the regulation of the expression and location of presynaptic GABA$_B$ receptors in glutamatergic and GABAergic neurons (*Luo et al., 2021*). We found that baclofen-induced inhibition of DAo was not different across genotypes (*Figure 4—figure supplement 2C*).

Because extracellular stimulation also recruits striatal cholinergic interneurons that greatly amplify DA axonal release through nicotinic receptor activation (*Liu et al., 2022*; *Rice and Cragg, 2004*; *Threlfell et al., 2012*; *Wang et al., 2014*; *Zhang and Sulzer, 2004*), we next isolated direct DA release from cholinergic regulation by blocking nicotinic receptors using dihydro-β-erythroidine hydrobromide (DHβE, 10 μM). Direct DA release isolated in this way was greatly reduced compared to release evoked in normal saline (*Figure 4—figure supplement 1K–L*), in line with previous work. Strikingly, peak DA release was reduced in DAT::NrxnsKO mice compared to WT in both the vSTR and dSTR (vSTR, Mann-Whitney test, p=0.0502 and dSTR, Mann-Whitney test, p=0.03) (*Figure 4I–J , and M–N*). The rate of DA recovery was unchanged (*Figure 4K and O*). Similarly, paired-pulse depression of this direct DA release was similar in DAT::NrxnsKO mice compared to WT mice (vSTR: unpaired t-test, 0.73 and dSTR: unpaired t-test, p=0.87) (*Figure 4L and P* and *Figure 4—figure supplement 1M–P*). Importantly, in these same recordings, prior to the addition of DHβE, we were able to recapitulate our previous observation of unaltered peak DAo and slowed rate of recovery in the vSTR of DAT::NrxnsKO mice compared to WT (DA release, dSTR: Mann-Whitney test, p=0.18 and tau, vSTR: Mann-Whitney test, p=0.0002) (*Figure 4—figure supplement 1E–J*).

Together these observations suggest that Nrxns regulate activity-dependent DAo through both direct and indirect mechanisms.

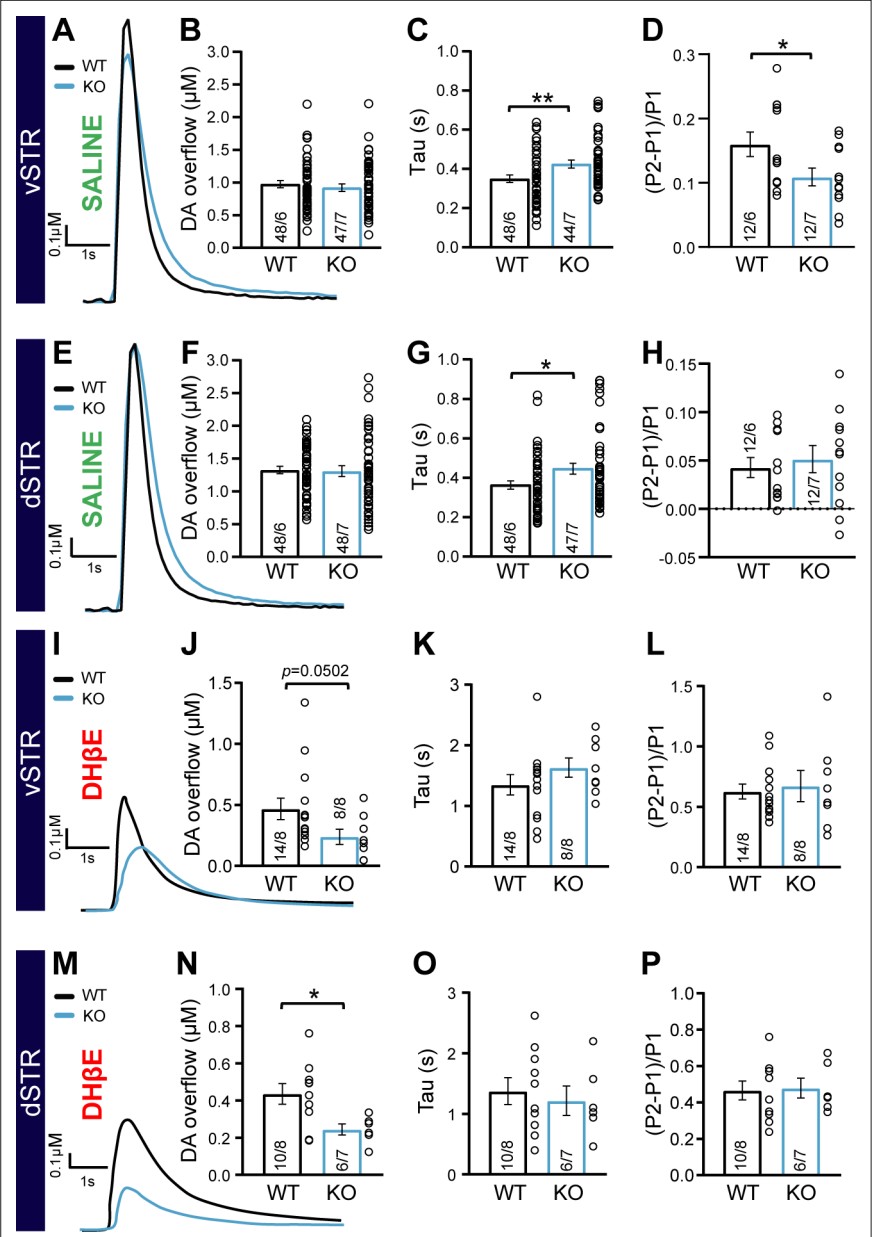

**Figure 4.** Impaired dopamine (DA) overflow in DAT::NrxnsKO mice. (**A**) Representative traces of electrically evoked DA overflow detected by fast-scan cyclic voltammetry in the ventral striatum, measured in slices prepared from DAT::NrxnsWT and DAT::NrxnsKO mice. (**B**) Bar graphs showing the average peak DA levels (µM) detected in the ventral striatum (WT = 0.98 ± 0.04 µM and KO = 0.98 ± 0.06 µM). (**C**) Evaluation of DA overflow kinetics in the ventral striatum estimated by quantifying tau (WT = 0.35 ± 0.02 and KO = 0.42 ± 0.02). (**D**) Short-term paired-pulse induced plasticity of DA overflow in ventral striatal slices, estimated by calculating (P2-P1/P1) with an inter-pulse interval of 100 ms. The low ratio values reflect the strong paired-pulse depression seen at such release sites in acute brain slices. (**E**) Representative traces of electrically evoked DA overflows detected by fast-scan cyclic voltammetry in the dorsal striatum. (**F**) Bar graphs showing the average peak DA levels (µM) detected in the dorsal striatum (WT = 1.33 ± 0.05 µM and KO = 1.35 ± 0.07 µM). (**G**) Evaluation of DA overflow kinetics in the dorsal striatum, estimated by quantifying tau (WT = 0.36 ± 0.02 s and KO = 0.45 ± 0.03 s). (**H**) Short-term paired-pulse induced plasticity of DA overflow in dorsal striatal slices, estimated by calculating (P2-P1/P1) with an inter-pulse interval at 100 ms. The low ratio values reflect the strong paired-pulse depression seen at such release sites in acute brain slices. (**I**) Representative traces of electrically evoked DA overflow detected by fast-scan cyclic voltammetry in the ventral striatum, measured in slices prepared from DAT::NrxnsWT and DAT::NrxnsKO mice in the presence of the nicotinic receptor antagonist DHßE. (**J**) Bar graphs showing the average peak DA levels (µM)

*Figure 4 continued on next page*

*Figure 4 continued*

detected in the ventral striatum (WT = 0.47 ± 0.09 µM and KO = 0.24 ± 0.06 µM). (**K**) Evaluation of DA overflow kinetics in the ventral striatum estimated by quantifying tau (WT = 1.35 ± 0.17 s and KO = 1.63 ± 0.16 s). (**L**) Short-term paired-pulse induced plasticity of DA overflow in ventral striatal slices, estimated by calculating (P2-P1/P1) with an inter-pulse interval of 100 ms. The low ratio values reflect the strong paired-pulse depression seen at such release sites in acute brain slices. (**M**) Representative traces of electrically evoked DA overflow detected by fast-scan cyclic voltammetry in the dorsal striatum in the presence of the nicotinic receptor antagonist DHßE. (**N**) Bar graphs showing the average peak DA levels (µM) detected in the dorsal striatum (WT = 0.43 ± 0.05 µM and KO = 0.24 ± 0.03 µM). (**O**) Evaluation of DA overflow kinetics in the dorsal striatum, estimated by quantifying tau (WT = 1.37 ± 0.22 s and KO = 1.21 ± 0.24 s). (**P**) Short-term paired-pulse induced plasticity of DA overflow in dorsal striatal slices, estimated by calculating (P2-P1/P1) with an inter-pulse interval at 100 ms. The low ratio values reflect the strong paired-pulse depression seen at such release sites in acute brain slices. Data are presented as mean ± SEM. Statistical analyses were performed with Student's t-tests (*p<0.05; **p<0.01).

The online version of this article includes the following source data and figure supplement(s) for figure 4:

**Source data 1.** Contains the primary data for *Figure 4* and *Figure 4—figure supplements 1 and 2*.

**Figure supplement 1.** Detection of activity-dependent dopamine (DA) overflow by fast-scan cyclic voltammetry (FSCV).

**Figure supplement 2.** No change in GABA_B receptor modulation of dopamine (DA) release after conditional deletion of all neurexins.

## GABA but not glutamate release by DA neurons is increased in the vSTR of DAT::NrxnsKO mice

Because DA neurons can also release GABA or glutamate at a subset of their axon terminals, we also examined whether loss of Nrxns in DA neurons alters GABA or glutamate-mediated synaptic currents evoked by optogenetic activation of DA neuron axons. A conditional AAV construct containing ChR2-EYFP was injected in the ventral mesencephalon and infected most of the neurons in the VTA and SNc of DAT::Nrxns mice (*Figure 5A–B* and *Figure 5—figure supplement 1*) to selectively measure optically stimulated synaptic responses in the medium spiny neurons (MSNs) innervated by the DA neuron axons. Whole-cell patch-clamp recordings in MSNs of the vSTR and dSTR in DAT::NrxnsWT and KO littermates were performed. We isolated GABA-mediated synaptic currents (inhibitory postsynaptic current [IPSC]) pharmacologically and IPSCs were evoked using brief blue light pulses (oIPSCs). oIPSCs were blocked by picrotoxin (50 µM), confirming their GABAergic nature (*Figure 5—figure supplement 2A*). Furthermore, oIPSCs were blocked after superfusion with the VMAT2 inhibitor reserpine (1 µM), in line with previous work showing that GABA release by DA neurons paradoxically requires VMAT2 (*Tritsch et al., 2014*; *Tritsch et al., 2012*; *Figure 5—figure supplement 2B*).

We found a significant increase in the amplitude of optically evoked IPSCs (oIPSCs) in MSNs recorded in vSTR slices prepared from DAT::NrxnsKO mice (*Figure 5C*; Mann-Whitney test, p=0.0014). Closer analysis of the kinetics of GABA-mediated oIPSCs in the vSTR revealed that the decay time constant (tau) was unchanged (*Figure 5D*; Mann-Whitney test, p=0.06) in DAT::NrxnsKO mice. The oIPSC delay and rise time were also unchanged (*Figure 5—figure supplement 2C–D*; unpaired t-tests p=0.49 and p=0.78). Thus, these results suggest that Nrxns are not influencing GABA receptor dynamics and the increase of oIPSC amplitude is likely due to a presynaptic effect.

As our manipulation affected the entire mesolimbic pathway, we also performed parallel optical stimulation and recordings in the dSTR. We observed no statistically significant changes in oIPSC amplitude in DAT::NrxnsKO mice in the dSTR (*Figure 5E*; Mann-Whitney test, p=0.36). oIPSC decay time constant was also unchanged (*Figure 5F*; Mann-Whitney test, p=0.63), as were the oIPSC delay and rise time (*Figure 5—figure supplement 2E–F*; unpaired t-tests p=0.85 and p=0.38). These results suggest that Nrxns act as regulators of GABA co-transmission by DA neurons in a region-specific manner. Further work will be required to identify the origin of this selectivity.

We also examined glutamate release by DA neurons in the vSTR and dSTR. We did not detect genotype differences in optically evoked excitatory postsynaptic currents (EPSCs) (*Figure 6A*; Mann-Whitney test, p=0.77, *Figure 6B*; Mann-Whitney test, p=0.92), which were otherwise blocked by the ionotropic glutamate receptor antagonists CNQX and AP5 (*Figure 6—figure supplement 1*). These results suggest that Nrxns do not have a major role in regulating glutamate release from DA neurons.

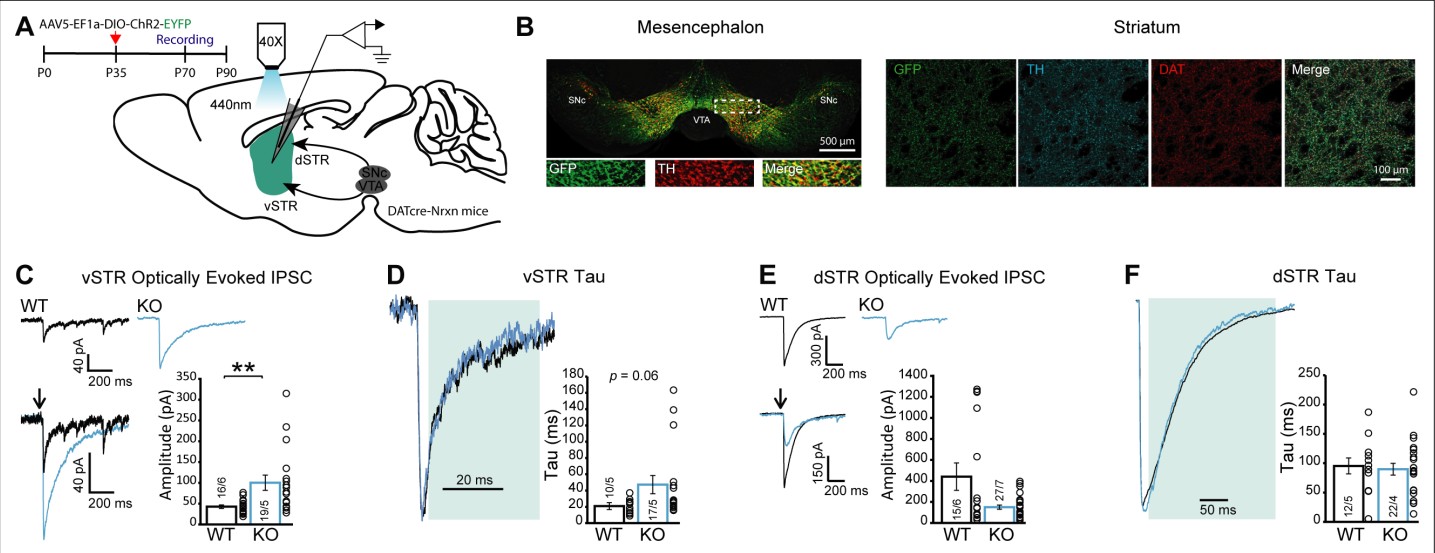

**Figure 5.** GABA release from dopamine (DA) neuron terminals in the ventral striatum is increased in DAT::NrxnsKO mice. (**A**) Experimental timeline and schematic for performing electrophysiological measurements from DAT::NrxnsKO and WT mice that were injected with AAV-EF1a-ChR2-EYFP in the ventral tegmental area (VTA) and substantia nigra pars compacta (SNc). (**B**) Representative image of virus expression in the mesencephalon (injection site) and striatum (projection) 6–8 weeks after stereotaxic viral injection. (**C**) Representative traces of optically evoked inhibitory postsynaptic currents (IPSCs) in the ventral striatum for WT and KO mice; summary plot showing a significant increase in average amplitude of optically evoked IPSCs for KO mice. (**D**) Representative traces of optically evoked IPSCs in the ventral striatum, shaded area represents the window used to calculate decay time constant; summary plot showing a trend toward an increase in decay time constant for KO mice. (**E**) Representative traces of optically evoked IPSCs in the dorsal striatum for WT and KO mice; summary plot showing no change in average amplitude of optically evoked IPSCs. (**F**) Representative traces of optically evoked IPSCs in the dorsal striatum, shaded area represents the window used to calculate decay time constant; summary plot showing no changes in decay time constant. Data are presented as mean ± SEM. Statistical analyses were performed with Mann-Whitney tests (**p<0.01).

The online version of this article includes the following source data and figure supplement(s) for figure 5:

**Source data 1.** Contains the primary data for *Figure 5* and *Figure 5—figure supplements 1 and 2*.

**Figure supplement 1.** Quantification of green fluorescent protein (GFP) signal in the striatum after viral transduction of dopamine neurons.

**Figure supplement 2.** GABA currents evoked from dopamine (DA) terminals in the ventral striatum (vSTR) are blocked by inhibiting vesicular monoamine transporter (VMAT2).

## Altered GABA uptake suggests a possible mechanism underlying increased GABA release in Nrxns KO mice

The region-specific increase of GABA release from DA axons in DAT::NrxnsKO mice, identified in our optogenetic experiments, could result from different mechanisms. One possibility is that loss of Nrxns induced differential adaptations in the expression of some of the postsynaptic partners of Nrxns, including NLs. However, using qRT-PCR, we did not detect major changes in expression of these genes in micro-dissected vSTR or dSTR, except for a small decrease in *collybistin* and *LRRTM3* mRNA in DAT::NrxnsKO mice (unpaired t-test, p=0.021 and p=0.014, respectively) (*Figure 7—figure supplement 1A–B*).

Another possibility is that the vesicular stores of GABA are increased in DAT::NrxnsKO mice. While DA neurons have the capacity to co-release GABA, it is well established that they do not synthesize it using glutamic acid decarboxylase and do not express the vesicular GABA transporter (*Tritsch et al., 2014*; *Tritsch et al., 2012*). Instead, they have been shown to use plasma membrane transporters to uptake GABA from the extracellular medium and VMAT2 to package it into synaptic vesicles (*Melani and Tritsch, 2022*; *Tritsch et al., 2014*; *Tritsch et al., 2012*). We therefore used a GABA uptake assay using primary DA neurons and quantified VMAT2 levels in the striatum.

GABA uptake was estimated using cultures prepared from DA neurons co-cultured with vSTR or dSTR neurons. Neurons were incubated with GABA (100 μM) for 2 hr, rapidly washed and fixed before quantification of GABA immunoreactivity in these neurons, evaluating the proportion of TH signal in DA axons covered by GABA immunoreactivity in comparison to a control group treated with $H_2O$ (*Figure 7A*). This treatment induced a robust increase in GABA immunoreactivity in DA neuron

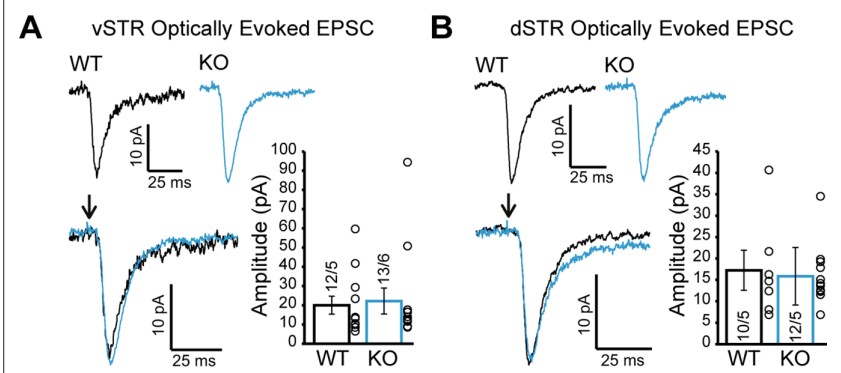

**Figure 6.** Glutamate release from dopamine (DA) axons in the ventral and dorsal striatum is unchanged in DAT::NrxnsKO mice. (**A**) Representative traces of optically evoked EPSCs in the ventral striatum for WT and KO mice; summary plot showing no differences in average peak amplitude for optically evoked EPSCs between WT and KO mice. (**B**) Representative traces of optically evoked EPSCs in the dorsal striatum for WT and KO mice; summary plot showing no differences in average peak amplitude for optically evoked EPSCs between WT and KO mice.

The online version of this article includes the following source data and figure supplement(s) for figure 6:

**Source data 1.** Contains the primary data for *Figure 6*.

**Figure supplement 1.** Glutamate-mediated synaptic currents evoked from dopamine (DA) terminals in the ventral striatum (vSTR) are blocked by the glutamate receptor antagonists AP5 and CNQX.

axons (*Figure 7B–D*). Interestingly, SNc DA neurons globally showed a larger GABA uptake compared to VTA DA neurons (*Figure 7E*; two-way ANOVA, main effect of region, $F_{(1, 76)}=53.35$, $p<0.001$). Furthermore, we observed a global increase of GABA uptake in DAT::NrxnsKO DA neurons compared to DAT::NrxnsWT DA neurons (two-way ANOVA, main effect of genotype, $F_{(1, 76)}=6.25$, $p=0.014$). This observation suggests that one possible mechanism underlying the increase in GABA release from DA neurons in the DAT::NrxnsKO is increased GABA uptake, leading to increased vesicular packaging and subsequent release. Additional work will be required to further test this hypothesis.

## Discussion

Since the initial discovery of Nrxns (*Ushkaryov et al., 1992*), multiple studies have explored the roles of these proteins in synapse formation, function, maintenance, and plasticity (*Südhof, 2017*). Most of these studies have been conducted on glutamatergic or GABAergic neurons, with no evaluation of their role in modulatory neurons such as DA, serotonin, norepinephrine, or acetylcholine neurons, whose connectivity is strikingly different and markedly less synaptic (*Ducrot et al., 2021*). We expected that new insights could be gained by studying the role of these trans-synaptic proteins in modulatory neurons. In the present study, we utilized the Cre-lox system by combining the triple conditional Nrxn mouse line (*Chen et al., 2017*) with a DAT-Cre mouse line to selectively delete Nrxns in DA neurons and examine the impact of this deletion using a combination of behavioral assessments, immunohisto-chemistry, electron microscopy, FSCV, and patch-clamp recordings of striatal MSNs. Considering that the *DAT* gene is turned on at around embryonic days 14–15, the gene deletion is expected to have happened at early stages of the establishment of DA neuron connectivity. More extensive changes in the basic connectivity of DA neurons could perhaps have been detected with an earlier KO. However, in the present work, we did not directly validate the precise moment at which Nrxns are removed from DA neurons.

We found that loss of Nrxns is associated with impaired DA neurotransmission in the brain of adult mice, as revealed by impaired amphetamine-induced locomotion, altered expression of key DA neuron markers, and a reduced activity-dependent DAo. However, the axonal ultrastructure of DA neuron terminals was unaltered. Patch-clamp recordings of GABA and glutamate release by DA neuron axons also revealed an unexpected increase of GABA co-release by DA neurons in the absence of Nrxns. Together these findings suggest that, although Nrxns may not be required for the

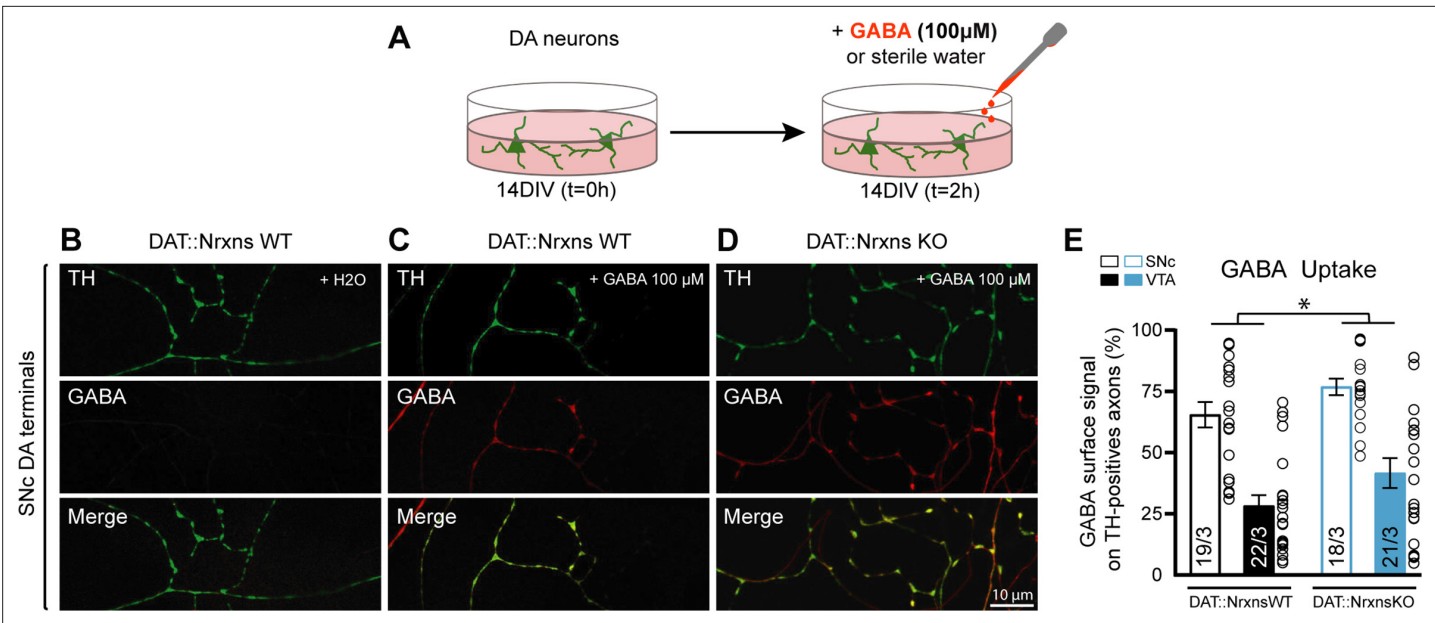

**Figure 7.** GABA uptake by cultured dopamine (DA) neurons is unchanged after conditional deletion of all neurexins (Nrxns). (**A**) Schematic representation of the experimental procedure for the GABA uptake assay in ventral tegmental area (VTA)-ventral striatum (vSTR) co-cultures or in substantia nigra pars compacta (SNc)-dorsal striatum (dSTR) co-cultures. (**B–D**) Immunocytochemistry of SNc DA neurons from DAT::NrxnsWT (**B and C**) and DAT::NrxnsKO mice (**D**) for tyrosine hydroxylase (TH, green) and gamma-aminobutyric acid (GABA, red). Experiments on VTA-vSTR co-cultures are not illustrated. (**E**) Summary graph representing the quantification of GABA immunoreactivity signal surface in TH-positive axons for VTA and SNc DA neurons from DAT::NrxnsWT and KO cultures. N=18–22 axonal fields from three different neuronal co-cultures. The number of observations represents the number of fields from TH-positive neurons examined. The star represents a significant overall genotype effect. For all analyses, plots represent the mean ± SEM. Statistical analyses were carried out by two-way ANOVAs followed by Tukey's multiple comparison test (*p<0.05).

The online version of this article includes the following source data and figure supplement(s) for figure 7:

**Source data 1.** Contains the primary data for *Figure 7* and *Figure 7—figure supplement 1*.

**Figure supplement 1.** Gene expression profile in target cells of dopamine (DA) neurons after conditional deletion of Nrxn123.

basic axonal development of DA neurons, they act as regulators of GABA and DA signaling in these neurons.

## Nrxns are not required for the basic morphological development of DA neuron release sites

Nrxns have been previously suggested to contribute to the development of synapses (*Aoto et al., 2015*; *Chen et al., 2017*; *Etherton et al., 2009*; *Li et al., 2007*; *Missler et al., 2003*). Here, we used electron microscopy to examine the ultrastructural characteristics of DA neuron axon terminals in the vSTR of DAT::NrxnsKO mice and DAT::NrxnsWT controls. Our observation of an absence of major structural changes in the DA neuron terminals after deletion of Nrxns is also in keeping with previous reports obtained with the single, double, or triple deletion of Nrxn in glutamatergic or GABAergic neurons (*Chen et al., 2017*; *Missler et al., 2003*). Another recent study using the triple Nrxn mice also reported no changes of synapse formation at the calyx of Held (*Luo et al., 2020*). We similarly conclude that Nrxns do not act as necessary drivers of axon terminal and synapse formation by DA neurons. In the present experiments, DA neuron terminals in the striatum were identified based on TH immunoreactivity, an approach that may miss some terminals containing low levels of this enzyme. Further immuno-EM experiments would however be required to examine glutamate-releasing terminals established by DA neurons and that can be identified by the presence of VGLUT2 (*Bérubé-Carrière et al., 2009*; *Dal Bo et al., 2004*; *El Mestikawy et al., 2011*; *Fortin et al., 2019*).

## DAo is altered in DAT::NrxnsKO mice

To obtain direct functional insight into the roles of Nrxns in dopaminergic neurotransmission, we performed FSCV recordings in the vSTR and dSTR, both under baseline conditions and after the

pharmacological blockade of nicotinic receptors. Under baseline conditions, peak electrically evoked DAo was similar in sections from DAT::NrxnsWT mice and DAT::NrxnsKO mice. However, in the presence of a nicotinic receptor antagonist, KO mice showed a clear reduction of peak DAo, suggesting that Nrxn123, although not playing an obligatory role in the DA release mechanism and in the initial formation and function of DA neuron varicosities, act as regulators of DA secretion. We also found slowed kinetics of recovery of extracellular DA in DAT::NrxnsKO mice compared to DAT::NrxnsWT mice. This could result either from reduced DAT function or from prolonged cholinergic amplification of DA neuron terminal activation. Arguing against the former and in favor of the later, we observed no significant change in the kinetics of DAo recovery when recordings were performed in the presence of the nicotinic antagonist DHβE. Previous work has suggested that Nrxns can regulate the localization of terminal nicotinic receptors in hippocampal neurons (*Cheng et al., 2009*), but this has never been examined in DA neurons. Our observation of altered DA release is particularly intriguing in the context of our finding that DAT::NrxnsKO mice show a robust impairment in amphetamine-induced locomotion. This reduced response to amphetamine could perhaps result from reduced DA stores in DA neuron terminals. However, this is not consistent with our observation of unchanged levels of TH and VMAT2 in the dSTR and increased VMAT2 in the vSTR. Amphetamine is well known to increase extracellular DA levels by impairing vesicular storage of DA and inducing reverse transport through the DAT. It is possible that Nrxns regulate the stability and function of the DAT in DA neuron axon terminals, perhaps through DAT's PDZ domain, and that in the absence of Nrxns, DAT function and positioning in terminals is impaired (*Sørensen et al., 2021*). It is thus possible that reduced DAT levels in the vSTR of Nrxn KO mice is implicated in the reduced locomotor response to amphetamine by limiting the extent of DA reverse transport. Further experiments would be needed to further examine this. It would also be interesting to shed further light into the significance of the increase in VMAT2 immunolabeled surface in the vSTR in the face of unchanged signal intensity. This could possibly reflect a broader distribution of the protein in axonal varicosities and/or an increase in the axonal domain of DA neurons, similar to a previous observation seen in conditional D2 KO mice (*Giguère et al., 2019*). Why we detected a decrease in peak DAo in the DAT::NrxnsKO mice only in the presence of a nicotinic antagonist remains unclear. One possibility is that electrical activation of cholinergic axonal release sites is increased in the DAT:NrxnsKO mice, thus compensating for the reduced primary DA release. This could be because loss of Nrxns from DA terminals trans-synaptically perturbs cholinergic terminals, in line with the work mentioned previously and suggesting that Nrxns can regulate the localization of terminal nicotinic receptors in some neurons (*Cheng et al., 2009*). Further work using optogenetic activation of striatal cholinergic neurons might help to test this hypothesis.

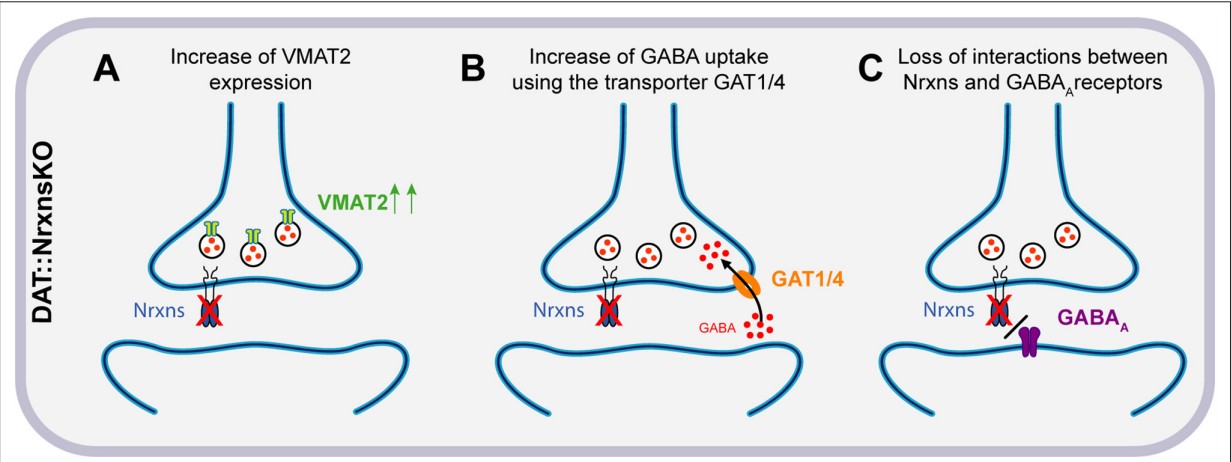

**Figure 8.** Hypothesized mechanisms of GABA co-transmission increase in DAT::NrxnsKO mice. (**A**) Illustration of the first hypothesis showing an increase of vesicular monoamine transporter (VMAT2) expression in DAT::NrxnsKO neurons, allowing increased vesicular GABA packaging. (**B**) Illustration of the second hypothesis showing an increase of GABA uptake through GAT1/4 in DAT::NrxnsKO neurons. (**C**) Illustration of the third hypothesis showing a possible loss of interaction between the presynaptic Nrxns and postsynaptic GABA_A receptors.

## Regulatory role of Nrxns in GABA release by DA neurons

One of the most intriguing observations in the present study is the region-specific increase of evoked GABA release from VTA DA neuron terminals in the vSTR. The origin of this selectivity is presently unclear but could arise from several possibilities including differential expression of Nrxn splice variants or their postsynaptic ligands and selective binding affinity in the dSTR or vSTR. Multiple studies over the past decade reported similar conclusions, where different Nrxn isoforms were proposed to regulate various aspects of synapse organization and function (*Ullrich et al., 1995*). Further studies would be needed to analyze the function of specific splice variants in each region. A possible hypothesis to explain our observations is that one or more Nrxns act as a repressor of GABA co-transmission and thus regulate the excitatory/inhibitory neurotransmission balance of the axonal domain of DA neurons. Indeed, previous work has shown that Nrxns physically and functionally interact with GABA$_A$ receptors and that overexpression of Nrxns decreases inhibitory but not excitatory synaptic strength (*Zhang et al., 2010*). These results are consistent with our observations showing that after removal of all Nrxns, GABA-mediated synaptic currents evoked by stimulation of DA neuron terminals are increased, perhaps in part through altered regulation of GABA$_A$ receptors (*Figure 8*). Further work, including rescue experiments, would be needed to test this hypothesis directly and determine which Nrxn is involved in this mechanism.

The regional specificity of our data raises intriguing questions regarding functional differences between the mesolimbic and nigrostriatal projections. We observed an increase in GABA-mediated oIPSCs only in the vSTR. Furthermore, basal GABA uptake was higher in SNc than VTA DA neurons and baseline oIPSC amplitude was higher in the dSTR than the vSTR. Together, these observations suggest that there may be intrinsic differences in the structure and function of GABA release sites in these two circuits.

While GABA release from DA neuron axon terminals was increased in the DAT::NrxnsKO mice, optically evoked synaptic glutamate release was unchanged. Nrxns therefore do not appear to act as major regulators of glutamate co-release by DA neurons. These findings highlight the complexity and the diversity of the role of Nrxns at synapses, in line with much recent work (*Chen et al., 2017*; *Luo et al., 2020*).

It would be interesting to complement our work by recording separately from striatal MSNs of the direct and indirect basal ganglia pathway (*Gerfen and Surmeier, 2011*), as some of the heterogeneity in our results on GABA and glutamate synaptic events may derive from differences in the roles of Nrxns in these two pathways. Previous work has shown that DA regulates tonic inhibition in striatal MSNs, and this regulation differs between D1 and D2 MSNs (*Janssen et al., 2009*). Given our results showing changes in GABA release from DA terminals, investigating tonic inhibition of D1 and D2 MSNs within the context of Nrxns deletion would be of interest for future work.

Mechanistically, the increase in GABA IPSCs we detected in DAT::NrxnsKO mice is likely due to an increase in GABA release from DA neurons. In keeping with this possibility, we detected an increase in GABA uptake by cultured DA neurons and an increase in VMAT2 levels in the vSTR. Together these two mechanisms could lead to increased GABA uptake by DA neuron terminals (possibly through GAT1/4) and increased GABA vesicular packaging through a VMAT2-dependent process (*Figure 8*). The lack of significant increase in VMAT2 in the dSTR could potentially explain why GABA release from DA neuron terminals was not increased in this part of the striatum even though an increase in GABA uptake was detected globally in both VTA and SNc neurons in the in vitro GABA uptake experiments. We also confirmed that oIPSCs in the vSTR were blocked almost entirely by reserpine, a VMAT2 inhibitor. The increase in oIPSC amplitude in the vSTR was not accompanied by changes in sIPSC amplitude. This further supports the idea that the changes observed in GABA-mediated currents resulted specifically from an increase in presynaptic GABA release from DA neurons (*Figure 8*). However, our results cannot formally exclude the implication of a postsynaptic mechanism. The trend for prolongation of oIPSCs that we detected in DAT::NrxnsKO mice may suggest that deletion of Nrxns from DA terminals increased the expression of postsynaptic GABA receptors. However, the non-significant trend toward increased decay tau of oIPSCs in vSTR does not fully account for the magnitude of increase in oIPSC amplitude. Thus, it is more likely that a presynaptic mechanism is the primary driver of the increased oIPSC amplitude observed in the vSTR. Further experiments will be required to identify the pre- or postsynaptic origin of this effect.

Together, our findings shed new light on the role of these cell adhesion molecules in DA neuron connectivity. We conclude that Nrxns are dispensable for the initial establishment of axon terminals and synapses by DA neurons but play a role in regulating both the kinetics of DAo and GABA release by DA neurons. Only a small subset of axon terminals established by DA neurons adopt a synaptic configuration. We conclude that the formation of such synaptic contacts must be regulated by other trans-synaptic proteins. Further studies are necessary to decipher the role of potential candidates including proteins from the leucocyte antigen receptor-protein tyrosine-phosphatases family. Our findings are globally compatible with those of a recent study evaluating the impact of deleting Nrxns from 5-HT neurons and showing reduced 5-HT release (*Cheung et al., 2023*). It would also be of interest to further extend this work by evaluating the role of Nrxns in noradrenergic and cholinergic neurons, populations of cells also developing dual connectivity and that are vulnerable in Parkinson's disease.

## Materials and methods

### Animals

All procedures involving animals and their care were conducted in accordance with the Guide to Care and Use of Experimental Animals of the Canadian Council on Animal Care. The experimental protocols (#21-113) were approved by the animal ethics committees of the Université de Montréal (CDEA). Housing was at a constant temperature (21°C) and humidity (60%), under a fixed 12 hr light/dark cycle with food and water available ad libitum.

### Generation of triple Nrxns cKO mice in DA neurons

All experiments were performed using mice generated by crossing DAT-IRES-Cre transgenic mice (Jackson Labs, B6.SJL-Slc6a3tm1.1 (Cre)Bkmn/J, strain 006660) with Nrxn123loxP mice (for details, see *Chen et al., 2017*). Briefly, cKO mice were produced as a result of CRE recombinase driving a selective excision of *Nrxn1*, *Nrxn2*, and *Nrxn3* genes in DA neurons and giving three different genotypes: DAT::NrxnsWT and DAT::NrxnsKO mice (for details, see *Figure 1—figure supplement 1*). The *Nrxn123*flox/flox mice were on a mixed Cd1/BL6 genetic background. The DAT-IRES-Cre mice were on a C57BL/6J genetic background. Except for culture experiments, only males were used.

### Genotyping

Mice were genotyped with a KAPA2G Fast HotStart DNA Polymerase kit (Kapa Biosystem). The following primers were used: *DAT-IRES-Cre*: Common 5' TGGCTGTTGTGTAAAGTGG3', wild-type reverse 5'GGACAGGGACATGGTTGACT 3' and knock-out reverse 5'-CCAAAAGACGGCAATATGGT -3', *Nrxn1* 5'-GTAGCCTGTTTACTGCAGTTCATT-3' and 5'-CAAGCACAGGATGTAATGGCCTT-3', *Nrxn2* 5'-CAGGGTAGGGTGTGGAATGAG-3' and 5'-GTTGAGCCTCACATCCCATTT-3', *Nrxn3* 5'-CCACACTTACTTCTGTGGATTGC-3' and 5'-CGTGGGGTATTTACGGATGAG-3'.

### Transmission electron microscopy

Following i.p. injection of ketamine (100 mg/kg) and xylazine (10 mg/kg), P70 mice were transcardially perfused with 50 ml of ice-cold sodium phosphate-buffered saline (PBS; 0.1 M; pH 7.4) followed by 150 ml of a mix composed of 4% paraformaldehyde (PFA) and 0.1% glutaraldehyde diluted in PBS. Dissected brains were extracted and post-fixed overnight in 4% PFA at 4°C. Mouse brains were cut with a vibratome (model VT1200 S; Leica, Germany) into 50-μm-thick transverse sections. For TH immunostaining, 50-μm-thick sections taken through the striatum (1.18 mm and 1.34 mm from bregma, according to the mouse brain atlas of Franklin and Paxinos, 1st edition) were rinsed with PBS and pre-incubated for 1 hr in a solution containing 2% normal goat serum and 0.5% gelatin diluted in PBS. Sections were then incubated overnight with a rabbit primary TH antibody (Millipore, catalogue no. AB152, 1/1000). Sections were rinsed and incubated during 2 hr with a goat anti-rabbit secondary antibody (1/500) and directly coupled to a peroxidase (Jackson, catalogue no. 111-035-003). The peroxidase activity was revealed by incubating sections for 5 min in a 0.025% solution of 3,3' diaminobenzidine tetrahydrochloride (Sigma-Aldrich, catalogue no. D5637) diluted in Tris-bufferred saline (TBS; 50 mM; pH 7.4), to which 0.005% of $H_2O_2$ was added. The reaction was stopped by several washes in TBS followed by phosphate buffer (50 mM; pH

7.4). At room temperature, sections were washed three times in ddH$_2$O and incubated for 1 hr in a solution composed of 1.5% potassium ferrocyanide and 2% osmium tetroxide (EMS) diluted in ddH$_2$O. After three rinses in ddH$_2$O, sections were incubated for 20 min in a filtered solution of 1% thiocarbohydrazide (Ted Pella) diluted in ddH$_2$O. Sections were then rinsed three times and incubated in 2% osmium tetroxide. After rinses in ddH$_2$O, sections were dehydrated in graded ethanol and propylene oxide and flat-embedded in Durcupan (catalogue no. 44611-14; Fluka, Buchs, Switzerland) for 72 hr at 60°C. Trapezoidal blocs of tissue from the vSTR were cut from the resin flat-embedded TH-immunostained sections. Each quadrangular pieces of tissue were glued on the tip of resin blocks and cut into 80 nm ultrathin sections with an ultramicrotome (model EM UC7, Leica). Ultrathin sections were collected on bare 150-mesh copper grids and examined under a TEM (Tecnai 12; Philips Electronic, Amsterdam, Netherlands) at 100 kV. Profiles of axon varicosities were readily identified as such by their diameter (larger than 0.25 µm) and their synaptic vesicles content. Using an integrated digital camera (MegaView II; Olympus, Münster, Germany), TH immunopositive axon varicosities were imaged randomly, at a working magnification of ×9000, by acquiring an image of every such profile encountered, until 50 or more showing a full contour and distinct content were available for analysis, in each mouse.

## Stereotaxic virus injections

DAT::NrxnsWT and DAT::NrxnsKO mice (P30–45) were anesthetized with isoflurane (Aerrane; Baxter, Deerfield, IL, USA) and fixed on a stereotaxic frame (Stoelting, Wood Dale, IL, USA). Fur on top of the head was trimmed, and the surgical area was disinfected with iodine alcohol. Throughout the entire procedure, eye gel (Lubrital, CDMV, Canada) was applied to the eyes and a heat pad was placed under the animal to keep it warm. Next, bupivacaine (5 mg/mL and 2 mg/kg, Marcaine; Hospira, Lake Forest, IL, USA) was subcutaneously injected at the surgical site, an incision of about 1 cm made with a scalpel blade and the cranium was exposed. Using a dental burr, one hole of 1 mm diameter was drilled above the site of injection (AP [anterior-posterior]; ML [medial-lateral]; DV [dorsal ventral], from bregma). The following injection coordinates were used: SNc/VTA (AP –3.0 mm; ML ±0.9 mm; DV –4.3 mm). Borosilicate pipettes were pulled using a Sutter Instrument, P-2000 puller, coupled to a 10 µl Hamilton syringe (Hamilton, 701 RN) using an RN adaptor (Hamilton, 55750-01) and the whole setup was filled with mineral oil. Using a Quintessential Stereotaxic Injector (Stoelting), solutions to be injected were pulled up in the glass pipette. For expression of ChR2 in DA neurons, 0.8 µl (VTA/SNc) of sterile NaCl containing 1.3×10$^{12}$ viral particles/mL of AAV5-EF1a-DIO-hChR2(H134R)-EYFP (UNC GTC Vector Core, NC, USA) was injected bilaterally. After each injection, the pipette was left in place for 5 min to allow diffusion and then slowly withdrawn. A second batch of mice were injected twice bilaterally (AP –2.8 mm; ML ±0.9 mm; DV –4.3 mm) and (AP –3.2 mm; ML ±1.5 mm; DV –4.2 mm) with a total of four 0.5 µl injections of the same viral preparation. This was done to improve the infection rate. Finally, the scalp skin was sutured and a subcutaneous injection of the anti-inflammatory drug carprofen (Rimadyl, 50 mg/mL at 5 mg/kg) was given. Animals recovered in their home cage and were closely monitored for 72 hr. A second dose of carprofen (5 mg/kg) was given if the mice showed evidence of pain. Finally, in the objective of validating the expression of ChR2 in DA neurons, some brains were used to quantify the EYFP signal in the striatum of P80 DAT::NrxnsWT and DAT::NrxnsKO mice. The results revealed very high overlap between the EYFP signal and the TH and DAT signals in DA neuron projections in both the vSTR and the dSTR (*Figure 5—figure supplement 1*).

## Electrophysiology and optogenetics

### Slice preparation

P70–80 mice were deeply anesthetized with isoflurane and quickly decapitated. Acute coronal slices (300 µm) were obtained using a vibrating blade microtome (Leica V1200S) in ice-cold *N*-methyl D-glucamine (NMDG) cutting solution: containing (in mM): 110 NMDG, 20 HEPES, 25 glucose, 30 NaHCO$_3$, 1.25 NaH$_2$PO$_4$, 2.5 KCl, 5 ascorbic acid, 3 Na-pyruvate, 2 thiourea, 10 MgSO$_4$-7 H$_2$O, 0.5 CaCl$_2$, 305–310 mOsm, pH 7.4. Slices equilibrated in a homemade chamber for 2–3 min (31°C) in the above solution and an additional 60 min in room temperature artificial cerebrospinal fluid (aCSF) containing (in mM): 120 NaCl, 26 NaHCO$_3$, 1 NaH$_2$PO$_4$, 2.5 KCl, 11 glucose, 1.3 MgSO$_4$-7 H$_2$O, and 2.5 CaCl$_2$ (290–300 mOsm, pH 7.4) before being transferred to a recording chamber.

## Whole-cell patch-clamp

Recordings were obtained from MSNs in the dorsal and vSTR. Striatal MSNs were visualized under infrared differential interference contrast (IR-DIC). Data were collected with a Multiclamp 700B amplifier, Digidata 1550B (Molecular Devices), and using Clampex 11 (pClamp; Molecular Devices, San Jose, CA, USA). All recordings were acquired in voltage clamp ($V_h$ = –70 mV) at 31°C and QX-314 (1 mM) was used in all internal solutions to internally block sodium channels. Whole-cell currents were acquired and sampled at 10 kHz with a low-pass Bessel filter set at 2 kHz and digitized at 10 kHz. For excitatory currents (EPSCs), the patch pipette was filled with internal solution containing (in mM): 135 CsMeSO$_4$, 8 CsCl, 10 HEPES, 0.25 EGTA, 10 phosphocreatine, 4 MgATP, and 0.3 NaGTP (295–305 mOsm, pH 7.2 with CsOH) and picrotoxin (50 µM) was added to aCSF. For IPSC, patch pipettes were filled with internal solution containing (in mM): 143 CsCl, 10 HEPES, 0.25 EGTA, 10 phosphocreatine, 4 MgATP, and 0.3 Na-2GTP (osmolarity 295–305, pH 7.2 with CsOH) and CNQX (10 µM) and AP5 (50 µM) were added to aCSF. All pipettes (3–4 MΩ) were pulled from borosilicate glass (Narishige PC-100). Patched cells were allowed a minimum of 3 min to stabilize following break-in and access resistance (Ra) was monitored throughout the recording and cells with an increase of >20% in Ra were discarded. Optically evoked synaptic currents were induced with 440 nm wavelength LED light delivered through a 40× objective lens (Olympus BX51WI) at 0.1 Hz (5 ms pulse) and light intensity was adjusted using an LLE-SOLA-SE2 controller (Lumencore).

## Immunohistochemistry on brain slices

DAT::Nrxn123WT and DAT::Nrxn123KO mice (P90) were anesthetized using pentobarbital NaCl solution (7 mg/mL) injected intraperitoneally and then perfused intracardially with 20 mL of PBS followed by 30 mL of 4% PFA. The brains were extracted, placed in PFA for 48 hr and then in a 30% sucrose solution for 48 hr. After this period, brains were frozen in –30°C isopentane for 1 min. Forty-µm-thick coronal sections were then cut using a cryostat (Leica CM1800) and placed in antifreeze solution at –20°C until used. For slice immunostaining, after a PBS wash, the tissue was permeabilized, non-specific binding sites were blocked and slices were incubated overnight with a rabbit anti-TH (1:1000, AB152, MilliporeSigma, USA), a mouse anti-TH (1:1000, Clone LNC1, MAB318, MilliporeSigma, USA), a rat anti-DAT (1:2000, MAB369; MilliporeSigma, USA), a rabbit anti-VMAT2 (1:2000, kindly provided by Dr. GW Miller, Columbia University) or a chicken anti-GFP (1:1000, GFP-1020; Aves Labs, USA) antibody. Primary antibodies were subsequently detected with rabbit, rat, or chicken Alexa Fluor-488-conjugated, 546-conjugated, and/or 647-conjugated secondary antibodies (1:500, 2 hr incubation; Invitrogen). Slices were mounted on charged microscope slides (Superfrost/Plus, Fisher Scientific, Canada) and stored at 4°C prior to image acquisition.

## Fast-scan cyclic voltammetry

### Recordings in basal conditions

DAT::NrxnsWT and DAT::NrxnsKO mice (P90–150) were used for FSCV recordings. The animals were anesthetized with halothane, quickly decapitated and the brain harvested. Next, the brain was submersed in ice-cold oxygenated aCSF containing (in mM): NaCl (125), KCl (2.5), KH$_2$PO$_4$ (0.3), NaHCO$_3$ (26), glucose (10), CaCl$_2$ (2.4), MgSO$_4$ (1.3), and coronal striatal/nucleus accumbens brain slices of 300 µm were prepared with a Leica VT1000S vibrating blade microtome. Once sliced, the tissue was transferred to oxygenated aCSF at room temperature and allowed to recover for at least 1 hr. For recordings, slices were put in a custom-made recording chamber superfused with aCSF at 1 mL/min and maintained at 32°C with a TC-324B single channel heater controller. All solutions were adjusted at pH 7.35–7.4, 300 mOsm/kg and saturated with 95% O$_2$-5% CO$_2$ at least 30 min prior to the experiment. Electrically evoked action potential-induced DAo was measured by FSCV using a 7 µm diameter carbon fiber electrode crafted as previously described (*Martel et al., 2011*) and placed into the tissue ~100 µm below the surface and a bipolar electrode (Plastics One, Roanoke, VA, USA) was placed ~200 µm away. The electrodes were calibrated with 1 µM DA in aCSF before and after each slice was recorded and the mean of the current values obtained was used to determine the amount of released DA. After use, electrodes were cleaned with isopropyl alcohol (Bioshop). The potential of the carbon fiber electrode was scanned at a rate of 300 V/s according to a 10 ms triangular voltage wave (–400 mV to 1000 mV vs Ag/AgCl) with a 100 ms sampling interval, using a headstage preamp (Axon Instruments, CV 203BU) and a Axopatch 200B amplifier (Axon Instruments, Union City, CA,

USA). Data were acquired using a digidata 1440a analog to digital board converter (Axon Instruments) connected to a personal computer using Clampex (Axon Instruments). Slices were left to stabilize for 20 min before any electrochemical recordings. Evaluation of DA release was achieved by sampling four different subregions of the dSTR and four different subregions of the vSTR (nucleus accumbens core and shell) using slices originating from +1.34 to +0.98 using bregma as a reference. After positioning of the bipolar stimulation and carbon fiber electrodes in the tissue, single pulses (400 µA, 1 ms) were applied to trigger DA release. After sampling of DA release, paired-pulse ratio experiments were conducted using one spot in the dSTR and one in the nucleus accumbens core. At each spot, a series of single pulses every 2 min for 10 min was collected as a baseline, followed by a three series of single-pulse stimuli intercalated with paired pulses (100 Hz) every 2 min (double pulse of 1 ms, 400 µA, with an inter-pulse interval of 100 ms).

### Recordings in the presence of a nicotinic receptor antagonist

After sampling DA release, one location of dSTR and one location of NAc core were selected. During this experiment, DHβE (Tocris), a nicotinic acetylcholine receptor blocker, was added to the aCSF (10 µM) to remove any amplification of DA release by striatal cholinergic interneurons. At each location, a series of single pulses were administered every 2 min for 10 min to evoke direct DA release and obtain a baseline. This was followed by three sequences of single-pulse stimuli intercalated with paired pulses (double pulses of 1 ms, 400 µA, inter-pulse interval of 100 ms), performed every 2 min. The paired-pulse ratio was calculated by subtracting the value of the peak DAo triggered by the single pulse (P1) by the value of the peak DAo triggered by the double pulse (P2), divided by the single-pulse peak DAo (P2-P1)/P1.

### Kinetic analysis

For all recording, peak DAo was considered to be the peak height of DA concentration, and DA recovery kinetics, assumed to follow Michaelis-Menten kinetics, were estimated from the time constant tau (time taken for the DAo to fall to 36.7% of its peak value), a metric not strongly influenced by peak signal amplitude, and calculated using a MATLAB script.

## Behavioral testing

Before behavioral experiments, mice were transferred from the colony and were housed with a maximum of four mice per cage. All mice were handled for 3 consecutive days prior to start of the different tests. All tests were performed in the same order as described below. The animals were tested between 10:00 AM and 4:30 PM.

### Rotarod

The accelerating rotarod task was used to assess motor coordination and learning. The apparatus consisted of five rotating rods separated by walls and elevated 30 cm from the ground. P60–70 mice were pre-trained on the rod (LE8205, Harvard Apparatus) 1 day before the recording to reach a stable performance. Mice were required to remain on the rod for 1 min at a constant speed of 4 rpm with a maximum of three attempts. For the first step of the rotarod testing protocol, the first day of the data acquisition, mice were tested on an accelerated rotation 4–40 rpm over a 10 min period for two sessions with an interval of 1 hr. The latency to fall was recorded. The same parameters were used on the second test day, but three sessions were performed. On the last day of data acquisition, the mice performed four sessions with the same previous parameters. A second protocol was also used, in which mice were tested with an accelerated rotation 4–40 rpm over a 2 min period for all sessions with an interval of 1 hr. Each trial per day was analyzed separately and compared between the genotypes.

### Locomotor activity and psychostimulant-induced locomotor activity

To evaluate motor behavior, mice were placed in cages (Omnitech Electronics, Inc; USA) designed for activity monitoring using an infrared actimeter (Superflex sensor version 4.6, Omnitech Electronics; 40×40×30cm$^3$) for 20 min. Next, 0.9% saline or the drug treatments were injected intraperitoneally (10 mL/kg) in a randomized order for the different genotypes. Horizontal activity was scored for 40 min following the injection. To evaluate psychostimulant-induced motor behaviors, the mice were

placed in the infrared actimeter cages (Superflex sensor version 4.6, Omnitech Electronics) for 20 min. Then, amphetamine was injected intraperitoneally at 5 mg/kg (Tocris, UK) or cocaine hydrochloride at 20 mg/kg (Medisca, cat# 53-21-4, Canada) in a randomized order for the different genotypes. The total distance (horizontal locomotor activity) was scored for 40 min following the injection.

## Pole test

The test was conducted with a homemade 48 cm metal rod of 1 cm diameter covered with adhesive tape to facilitate traction, placed in a cage. Eight-week-old DAT::NrxnsWT and DAT::NrxnsKO mice were positioned head-up at the top of the pole and the time required to turn (t-turn) and climb down completely was recorded.

## Sucrose preference test

Mice were tested for preference of a 2% sucrose solution, using a two-bottle choice procedure. Subjects were housed one per cage all of the test (5 days). Mice were first conditioned with two bottles of water during two days. Then mice were given two bottles, one sucrose (2%) and one of tap water. Every 24 hrs, the amount of sucrose and water consumed was evaluated. To prevent potential location preference of drinking, the position of the bottles was changed every 24 hrs. The preference for the sucrose solution was calculated as the percentage of sucrose solution ingested relative to the total intake.

## Reverse transcription quantitative polymerase chain reaction

We used reverse transcription quantitative polymerase chain reaction (RT-qPCR) to quantify the amount of mRNA encoding the following genes: gephyrin (*Gphn*), collybistin (*Arhgef 9*), GABA-A receptor (*Gabra1*), NLs 1, 2, and 3 (*Nlgn1, -2, and -3*), latrophilins 1, 2 and 3 (*Lphn1, -2, and -3*), LRRTMs 1, 2, 3, 4 (*LRRTM1, -2, -3, and -4*), D1R (*DRD1*) and D2R (*DRD2*) in striatal brain tissue from P80 DAT::Nrxn123WT and DAT::Nrxn123 KO mice. The brains were quickly harvested and the vSTR and dSTR were micro-dissected and homogenized in 500 µL of Trizol. For both, presynaptic and postsynaptic compartment, RNA extraction was performed using RNeasy Mini Kit (QIAGEN, Canada) according to

**Table 2.** qPCR primers for ventral and dorsal striatum.

| Gene | Oligo forward | Oligo reverse | Reference sequences |
|------|---------------|---------------|---------------------|
| *Gphn* | cctcgcccagaataccac | gacggctgctcatctgattac | NM_145965.2, NM_172952.3 |
| *Arhgef9* | tgagaaaagcttctaaacagaaagg | gtactggccctggtttaacg | NM_001033329.3 |
| *Gabra1* | cgatcctctctcccacactt | tttcttcatcacgggcttg | NM_010250.5 |
| *Nlgn1* | ctatcggcttgggggtacttg | caaggagcccgtagtttcct | NM_138666.3, NM_001163387.1 |
| *Nlgn2* | gaggaaaggggggaatctctg | ggccgtgggaaggtaagt | NM_198862.2 |
| *Nlgn3* | gaagggagggctccaagat | ggtccttctccttggtctgat | NM_172932.4 |
| *Adgrl1* | cagtacgactgtgtcccttacatc | cagactgatgctctgactcatgt | NM_181039.2 |
| *Adgrl2* | gagctgaagccgagtgagaa | cctgcatgtcttctctcgttt | NM_001081298.1 |
| *Adgrl3* | aacaacctccttcagccaca | cgcagttgatcacttgtcgt | NM_001347369.1 |
| *Lrrtm1* | cgccctgcatataattagcc | gaagcgctgggtcagaaa | NM_028880.3 |
| *Lrrtm2* | gtagggacaaaaacctgtttgatt | aagtaggaagccagttgtggtc | NM_178005.4 |
| *Lrrtm3* | gaccctgcacctatagcaaatc | tgccagaaaggttgacacat | NM_178678.4 |
| *Lrrtm4* | gccatgattctcctggtgat | tgagtgctgttggagttgtttc | NM_001134743.1 |
| *Drd1* | aggttgagcaggacatacgc | tggctacggggatgtaaaag | NM_010076.3 |
| *Drd2* | gatgcttgccattgttcttg | attcaggatgtgcgtgatga | NM_010077.2 |
| *Gapdh* | tgtccgtcgtggatctgac | cctgcttcaccaccttcttg | NM_008084.2 |
| *Actb* | aaggccaaccgtgaaaagat | gtggtacgaccagaggcatac | NM_007393.3 |

the manufacturer's instructions. RNA integrity was validated using a Bioanalyzer 2100 (Agilent). Total RNA was treated with DNase and reverse transcribed using the Maxima First Strand cDNA synthesis kit with dsDNase (Thermo Fisher Scientific). Gene expression was determined using assays designed with the Universal Probe Library from Roche (https://www.universalprobelibrary.com). For each qPCR assay, a standard curve was performed to ensure that the efficiency of the assay was between 90% and 110%. The primers used are listed in *Table 2*. The Quant Studio qPCR instrument (Thermo Fisher Scientific) was used to detect the amplification level. Relative expression comparison (RQ = $2^{-\Delta\Delta CT}$) was calculated using the Expression Suite software (Thermo Fisher Scientific), using GAPDH and β-Actin as an endogenous control.

## Transcriptome analysis

Transgenic mice expressing the *eGFP* gene in monoaminergic neurons under control of the TH promoter (TH-GFP mice) were used to manually dissect the SNc and VTA and isolate DA neurons. P0–2 mice were cryo-anesthetized and decapitated for tissue collection. As described previously (*Ducrot et al., 2021*; *Fulton et al., 2011*; *Mendez et al., 2008*), freshly dissociated cells from the VTA were prepared and GFP-positive neurons were purified by FACS and directly collected in Trizol (QIAGEN). RNA extraction was performed with the RNeasy Mini kit (QIAGEN) according to the manufacturer's instructions. The concentration and the purity of the RNA from DA neurons was determined using Qubit (Thermo Scientific) and quality was assessed with the 2100 Bioanalyzer (Agilent Technologies). Transcriptome libraries from three independent samples of VTA or SNc DA neurons were generated using the Truseq RNA Stranded (Illumina) using a poly-A selection. The precision of the dissection of VTA and SNc tissues was validated by the strong enrichment of genes known to be expressed at higher levels in SNc (*Sox6*) or the VTA (*Slc17a6*, *Calb1*). Sequencing was performed on the HiSeq 2000 (Illumina), obtaining around 15 M paired-end 100 bp reads per sample. Statistical analysis was performed with DESeq2 software by using the Wald test as described previously (*Love et al., 2014*; Genome Biology). Values are presented as FKPM (fragments per kilobase of transcript per million fragments mapped) values. The DESeq2 method uses raw read counts that are internally normalized (https://chipster.csc.fi/manual/deseq2.html).

## Primary cell culture

For GABA uptake experiments, we used primary neuronal cultures. Briefly, postnatal day o-3 (P0–3) mice were cryoanesthetized, decapitated, and used for co-cultures as previously described (*Ducrot et al., 2021*; *Fasano et al., 2008*). Primary VTA or SNc DA neurons were separately dissected from Nrxn123 KO or Nrxn123 WT pups and co-cultured with vSTR and dSTR neurons, respectively, from Nrxn123 KO or Nrxn123 WT pups. Neurons were seeded on a monolayer of cortical astrocytes grown on collagen/poly-L-lysine-coated glass coverslips. All cultures were incubated at 37°C in 5% $CO_2$ and maintained in 2/3 of Neurobasal medium, enriched with 1% penicillin/streptomycin, 1% Glutamax, 2% B-27 supplement, and 5% fetal bovine serum (Invitrogen) plus 1/3 of minimum essential medium enriched with 1% penicillin/streptomycin, 1% Glutamax, 20 mM glucose, 1 mM sodium pyruvate, and 100 μL of MITO+ serum extender. All primary neuronal co-cultures were used at 14DIV.

## Immunocytochemistry on cell cultures

Cultures used for GABA uptake experiments were fixed at 14DIV with 4% PFA (in PBS, pH 7.4), permeabilized with 0,1% triton X-100 during 20-min, and nonspecific binding sites were blocked with 10% bovine serum albumin during 10 min. Primary antibodies used were: mouse anti-tyrosine hydroxylase (TH) (1:2000; Sigma) and rabbit anti-GABA (1:500, Millipore-Sigma). These were subsequently detected using Alexa Fluor-488-conjugated, and Alexa Fluor-647-conjugated secondary antibodies (1:500, Invitrogen). Coverslips were mounted on microscope slides (Fisher Scientific, Canada) and stored at 4°C prior to confocval image acquisition.

## Image acquisition with confocal microscopy

All in vitro fluorescence imaging quantification analyses were performed on images acquired using an Olympus Fluoview FV1000 point-scanning confocal microscope (Olympus, Tokyo, Japan). Images

were scanned sequentially to prevent non-specific bleed-through signal using 488, 546, and 647 nm laser excitation and a 60× (NA 1:42) oil immersion objective.

For the immunohistochemical characterization of brain tissue obtained from DAT::NrxnsWT and DAT::NrxnsKO mice, surface and intensity for each signal were measured in a series of three different striatal sections ranging from bregma +0.74 to bregma –0.82 mm, with a total of 14 different spots for each hemisphere. All quantifications were performed by using a macro developed in-house.

## Image analysis

All image quantification was performed using ImageJ (National Institutes of Health [NIH]) software. A background correction was first applied at the same level for every image analyzed before any quantification. A macro developed in-house was used to perform all quantifications.

## Statistics

Data are represented throughout as mean ± SEM. Statistically significant differences were analyzed using Student's t test, Mann-Whitney test, one-way repeated measures ANOVA or two-way ANOVA with Tukey's or Sidak's multiple comparison test ($*p < 0.05$; $**p < 0.01$; $***p < 0.001$; $****p < 0.0001$).

## Acknowledgements

We would like to thank Dr. G Miller for kindly provided VMAT2 antibody, Willemieke Kouwenhoven and Alex Tchung for their help with some experiments and for data analysis, the IRIC genomic platform for qRT-PCR analysis, CERVO Québec electron microscopy platform, and CA Maurice for his strong support. This work was funded by Canadian Institutes of Health Research (CIHR, grant MOP106556) to L-ET and University of California, Irvine (School of Medicine, GF15247) to LYC. CD received a graduate student award from Fond de Recherche en Santé du Québec (FRSQ).

## Additional information

### Funding

| Funder | Grant reference number | Author |
| --- | --- | --- |
| Canadian Institutes of Health Research | MOP106556 | Louis-Eric Trudeau |
| University of California Irvine, School of Medicine | GF15247 | Lulu Y Chen |

The funders had no role in study design, data collection and interpretation, or the decision to submit the work for publication.

### Author contributions

Charles Ducrot, Data curation, Formal analysis, Investigation, Methodology, Writing – original draft, Writing – review and editing; Gregory de Carvalho, Formal analysis, Validation, Investigation, Writing – review and editing; Benoît Delignat-Lavaud, Constantin VL Delmas, Priyabrata Halder, Sriparna Mukherjee, Formal analysis, Investigation, Methodology; Nicolas Giguère, Data curation, Formal analysis, Investigation, Methodology; Consiglia Pacelli, Investigation, Methodology; Marie-Josée Bourque, Data curation, Methodology; Martin Parent, Data curation, Supervision, Validation, Methodology, Writing – review and editing; Lulu Y Chen, Conceptualization, Resources, Data curation, Supervision, Validation, Writing – original draft, Project administration, Writing – review and editing; Louis-Eric Trudeau, Conceptualization, Resources, Data curation, Supervision, Funding acquisition, Validation, Writing – original draft, Project administration, Writing – review and editing

### Author ORCIDs

Charles Ducrot  http://orcid.org/0000-0002-5451-1610
Gregory de Carvalho  http://orcid.org/0000-0002-9179-7697
Benoît Delignat-Lavaud  http://orcid.org/0000-0003-4680-9115
Consiglia Pacelli  http://orcid.org/0000-0003-4915-5823

Sriparna Mukherjee http://orcid.org/0000-0002-1299-4343
Martin Parent http://orcid.org/0000-0002-0868-1010
Lulu Y Chen http://orcid.org/0000-0002-8873-3481
Louis-Eric Trudeau http://orcid.org/0000-0003-4684-1377

### Ethics

All procedures involving animals and their care were conducted in accordance with the Guide to care and use of Experimental Animals of the Canadian Council on Animal Care. The experimental protocols (#21-113) were approved by the animal ethics committees of the Université de Montréal (CDEA).

### Decision letter and Author response

Decision letter https://doi.org/10.7554/eLife.87902.sa1
Author response https://doi.org/10.7554/eLife.87902.sa2

## Additional files

### Supplementary files
• MDAR checklist

### Data availability
All primary data are provided in the source data files accompanying the manuscript.

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
