## [Editor Report]

In this study, the authors selectively delete the three main genes encoding neurexins from dopamine neurons in mice. The authors find that while dopamine axonal architecture and synaptic ultrastructure are generally unaffected by loss of neurexins, there are changes in dopamine reuptake, amphetamine-induced locomotion, and GABA co-release, and notably, these changes are region specific, with most of the effects observed in the ventral striatum. The results are solid, and these findings are valuable and useful, providing new information regarding the potential roles of neurexins in regulating dopamine neuron output.

---

## [Decision Letter]

**Decision letter after peer review:**

[Editors’ note: the authors submitted for reconsideration following the decision after peer review. What follows is the decision letter after the first round of review.]

Thank you for submitting the paper "Conditional deletion of neurexins dysregulates neurotransmission from dopamine neurons" for consideration by *eLife*. Your article has been reviewed by 3 peer reviewers, and the evaluation has been overseen by a Reviewing Editor and a Senior Editor.

Comments to the Authors:

We are sorry to say that, after consultation with the reviewers, we have decided that this work in its current form will not be considered further for publication by *eLife*.

The reviewers acknowledge the overall significance of the work, though they nevertheless all have several important comments. In particular, they feel that at present the study is somewhat underpowered because of the low N numbers in several experiments and that it would be important to have more rigorous controls. We discussed these points among the reviewers, and we take these points seriously and have the opinion that they need to be carefully experimentally addressed to ensure rigor.

However, if you can fully address the reviewer's concerns, we would be happy to encourage you to submit a revised manuscript as a new submission. We hope that you will find the reviewers' comments to be constructive and if you have any questions about the reviews please let us know.

*Reviewer #1 (Recommendations for the authors):*

By generating a mouse model in which neurexins (Nrxns) are selectively deleted from dopamine (DA) neurons, Ducrot et al. explore the role of these presynaptic adhesion proteins in neurotransmission from DA neurons. They show that mice in which Nrxn 1, 2, and 3 are deleted from DA neurons display a significant reduction in amphetamine-induced locomotion, but no alterations in basal movement, motor coordination/learning on the rotarod, and cocaine-induced activity. Immunohistochemistry revealed that Nrxns KO mice show an increase in VMAT2 and a reduction in DAT-positive (but not intensity) in ventral striatum (vStr), but not dorsal striatum (dStr). Using fast-scan cyclic voltammetry, the authors reveal intact DA release but a reduced rate of DA uptake in vStr but not dStr. Moreover, the authors find that GABA co-release from DA neuron is increased in vStr of Nrxns KO mice, while glutamate transmission is not affected.

Major Strengths:

This study combines an impressive array of experimental methods to characterize these triple conditional knockout mice, including behavioral assays for basic locomotor function, immunohistochemistry of presynaptic DA markers, electron microscopy of DA synapse ultrastructure, fast-scan cyclic voltammetry of DA release, and whole-cell electrophysiology to monitor GABA and glutamate co-release. In most cases, analyses are applied to ventral and dorsal striatal regions for a comprehensive characterization of synaptic function from mesolimbic and nigrostriatal dopaminergic neurons.

Major Weaknesses:

This study is mainly concerned with providing a list of differences between control and knockout mice. While the number of assays used is impressive, the study does not deeply investigate any of the reported phenotypes, some of which are quite small in magnitude and suffer from low numbers of experimental observations. It is for instance unclear how the immunohistochemical findings relate to the behavior or voltammetry/electrophysiological data, why deficits are limited to the ventral striatum, or what the relative contributions to neurexins 1, 2, or 3 to the many phenotypes reported here are. Overall, this descriptive study is limited in the conclusions it can make, falling short of markedly increasing our understanding of the effects of neurexin 1, 2, or 3 on synaptic function.

Specific points:

1. It is not clear which neurexins midbrain dopamine neurons express, and whether differences exist between VTA and SNc neurons that may explain the lack of phenotype in SNc neurons? The authors first need to confirm which Neurexin(s) is expressed in DA neurons and that they are effectively lost from DA neurons in the knockout.

2. It is not clear how immunohistochemical changes in DAT or VMAT2 'surface' relate to DA neurotransmission, especially when more classical measures of protein abundance like intensity are not changed. For example, the authors suggest the decrease in DAT 'surface' may be related to the slower time constant of DA uptake observed with fast-scan cyclic voltammetry, but slowing of DA uptake is not accompanied by changes in DAT 'surface' in the dorsal striatum. Similarly, it is unclear why an increase in VMAT2 'surface' does not translate into greater DA release. Lastly, since the authors report a GABA co-release phenotype, the authors should complement their immunohistochemical analyses with protein targets related to GABA co-release. Together, they will help make sense of the phenotypes observed by suggesting mechanisms.

3. The authors report a potentiation of GABA co-release and a slowing down of IPSC decay kinetics in the vStr of Nrxn KO mice compared to WT, while dStr responses are unaffected. However, both data sets suffer from low Ns with a handful of outliers that may significantly skew population averages and confound interpretation. For example, Figures 5C and D each contain 3 data points that are significantly larger than all others. Are they from the same mouse, which may have had better virally-mediated ChR2 expression than the rest? The same is true for Figure 5E in dStr. Would the findings in vStr and dStr stand without these outliers? Given the observed variability of IPSC response magnitude when ChR2 is expressed virally, any conclusion regarding GABA co-release would need to be bolstered with larger Ns across more animals.

4. In Figure 7, the authors conclude that GABA uptake is much higher in Nrxns KO mice compared to WT mice. However, this result arises from a main effect of genotype following a 2-Way ANOVA, which pools together SNC and VTA data. Is a statistically significant increase still seen when simply comparing KO vs WT mice in SNc cultures, and separately KO vs. Wt mice in VTA cultures? The differences appear minor and are unlikely to explain the IPSC phenotype observed vStr but not dStr in Figure 5. Wouldn't an increase in GABA uptake speed up the decay time constant? How can the authors rule out the increase in VMAT2 'surface' underlying the latter?

5. DA Voltammetry is performed using electrical stimulation, which preferentially stimulates cholinergic interneurons and drives DA release indirectly via nicotinic acetylcholine receptors. This mechanism complicates analyses of DA release probability. The authors should repeat these analyses in the presence of DHBE or other nAChR antagonist to evoke DA release from DA axons directly. Only then can the release probability of DA from DA axons be directly evaluated.

1. Figure 2: In order to better appreciate the differences between WT and cKO in VMAT2 and Dat expression, the authors should provide more informative and representative images, possibly with higher contrast. For example, by looking at the images shown in Figure 2B, it looks like VMAT2 is fainter in dStr of WT mice than in cKO. Also in Figure 2D, Dat signal seems much lower in cKO. In addition, there is a typo in the y-axis of Figure 2I as I am assuming it should read "DAT signal surface".

2. Methodological details explaining the immunohistochemistry is quantified are missing.

3. Page 10, line5, the authors say "We focused on terminals in the vStr because this area contains DA neuron release sites for both glutamate and GABA in addition to those for DA". This statement is misleading as it suggests that both glutamate and GABA from DA neurons are only released in vStr.

4. There is a typo on page 22, line 14: "regulators if glutamate co-release".

5. Please show traces of paired-pulse depression in Figure 4 or in supplemental materials.

6. The increase in sIPSC frequency in vSTr of Nrxns cKO mice is not supported by an appropriate hypothesis. Although in the Discussion the authors state that "our observation of an increase in sIPSC frequency in the vSTR of KO mice could potentially result from alterations in DA neuron projections to non-striatal regions", no experiments are performed to support this, and the findings have little to no bearings on the rest of the study. I suggest removing these data.

*Reviewer #2 (Recommendations for the authors):*

The goal of this study was to determine how deletion of the genes encoding the neurexin family of trans-synaptic adhesion molecules (Nrxns), impacts neurotransmitter release from dopamine neurons. The authors used multiple approaches to provide a comprehensive overview of the consequences of Nrxn loss on the output of the dopaminergic system. Specifically, they show that while overall axonal architecture and dopamine release are unaffected by Nrxn loss, there are alterations in DAT function that impair dopamine re-uptake in the striatum. This leads to altered locomotor activation in response to amphetamine without affecting baseline motor function.

In addition to measuring dopamine, this study assesses GABA and glutamate release, which are co-transmitters in dopamine neurons. The authors make the interesting observation that GABA release from dopamine neurons is enhanced by loss of Nrxns, while glutamate release is unaltered. This potentiation of GABA release is notably region-specific and only occurs in the ventral but not dorsal striatum. The study goes on to show that GABA uptake from cultured Nrxn knock-out dopamine neurons is enhanced, suggesting an intriguing mechanism by which GABA signaling may be potentiated. Overall, this study provides the first look at how disruption of Nrxns affects dopamine neuron output, as these molecules have until now been studied on glutamatergic or GABA-ergic neurons, with complex effects on synaptic development and function.

The manuscript is well-written and the data generally support the conclusions (although see limitations below). The discussion of the data is balanced and the discussion highlights areas that warrant further investigation. The data is of good quality and the assays used represent the standard in the field and cover an array of dopamine neuron properties (e.g cellular morphology, subcellular ultrastructure, dopamine release & re-uptake, GABA, and glutamate release, motor behavior, and drug-induced behavior). The use of the GABA uptake assay in dopamine neurons is particularly innovative and compelling, especially as dopamine neurons are challenging to culture.

There are a few limitations of the study. Specifically, it is unclear whether DAT-Cre positive mice were used as controls or whether the controls were DAT-Cre negative. The presence of Cre reduces the expression of endogenous DAT in DAT-Cre mice, which alters dopamine release and behavior. Cre expression, therefore, needs to be present in all groups for an accurate comparison. If Cre+ controls were used, it would be helpful for the authors to describe their breeding scheme. If the Nrxn floxed/DAT-Cre+ mice were maintained as a separate line from Nrxn WT/DAT-Cre+ mice, this should be noted.

For the recordings of GABA and glutamate release within the striatum, unidentified striatal neurons (SPNs) were recorded. The two main types of SPNs can differ significantly in their intrinsic and synaptic properties; therefore, a more rigorous approach would be to identify and record these neurons separately. In addition, some of the physiology experiments use a relatively small number of neurons (<10). Given that the ChR2 is delivered virally and there could be significant differences in viral transduction between mice, a larger sample size (or control for viral expression) would be needed to strengthen the conclusions. The FSCV experiments use electrical stimulation but the effects of acetylcholine release on dopamine transmission have not been controlled for. It would be helpful to repeat these experiments using ChR2 stimulation or with cholinergic blockers to exclude any contribution of altered cholinergic signaling to the observed phenotypes. Finally, while this study provides an important characterization of the consequences of Nrxn loss, it does not delve into the mechanistic basis of these changes, which would be an interesting area for future investigation. Specifically, it would be interesting to know whether GABA membrane transporter expression (Gat) is altered in Nrxn KO dopamine neurons and how Nrxns interact with DAT.

Specific points:

1) It is not clear whether the control mice in this study were positive for DAT-Cre. i.e. were the controls WT for all Nrxns and DAT-Cre+? This is important as DAT expression and function are reduced in DAT-Cre+ mice, which affects dopamine release, reuptake, and basic locomotor behavior (e.g. see PMIDs: 16865686 and 33979604). This could be a confound if the Nrxn KO mice were DAT-Cre positive but the controls were DAT-Cre negative. If Cre+ controls were used, it would be helpful for the authors to describe their breeding scheme. If the Nrxn floxed/DAT-Cre+ mice were maintained as a separate line from Nrxn WT/DAT-Cre+ mice, this should be noted. It seems that it would be very challenging to generate littermate mice that were homozygous floxed or homozygous WT for all three Nrxn genes and also DAT-Cre positive.

2) The authors suggest that the sucrose preference task measures motivation. My understanding is that this test is a measure of hedonia/anhedonia rather than purely "motivation". The authors may want to revise the text here or consider a more classical test of motivation, e.g. lever pressing progressive-ratio test.

3) For many of the graphs, the authors have only shown error bars in one direction. It would be more rigorous to show the error bars in both directions. This allows easy assessment of whether the error bars overlap (typically denoting a lack of a significant difference between groups).

4) The authors should provide example FSCV traces for the paired-pulse experiments in Figure 4. In addition, the authors should control for any potential contribution of cholinergic alterations on dopamine release and reuptake. This could be done by including cholinergic blockers or by using ChR2 to evoke dopamine release.

5) The physiology results suggest an interesting increase in GABA but not glutamate release from dopamine neurons. These conclusions are based on a relatively small sample size for some experiments (<10 neurons) and there could be considerable variability in viral transduction across mice. It would strengthen the conclusions if more "n" could be added to these experiments.

6) For Figure 5, it would be helpful to show a higher magnification image of EYFP expression in the midbrain – to show that it is exclusively expressed in TH positive dopamine neurons and not neighboring GABA or glutamatergic neurons.

7) The GABA uptake experiments in Figure 7 are an innovative approach to assessing the potential mechanisms for increased GABA-ergic transmission from DA neurons. The authors mention that DA neurons were co-cultured with SPNs. How is GABA uptake affected in SPNs? i.e. We would assume that GABA uptake should only be enhanced in DA neurons from DAT-Nrxn KO, not the co-cultured SPNs. This would provide strong evidence that changes in the DA neurons themselves are responsible for enhanced oIPSCs.

8) How were the statistical comparisons done for Figure 7? From the graph, it is unclear which groups are statistically different. This is important as the effect size is fairly small.

9) Given the potential mechanism shown in Figure 7, it would be very interesting to know whether the expression of Gat1 is altered in DAT-Nrxn KO neurons, although it is understood that this may be beyond the scope of this study.

*Reviewer #3 (Recommendations for the authors):*

Overall, this is an interesting manuscript that uses established techniques to describe the impact of pan-Neurexin deletion on the cell-autonomous function of dopamine neurons in the mouse brain. Using cell type-specific conditional gene deletion, they find that Neurexins play a role in maintaining aspects of DA neurotransmission, particularly within the ventral striatum. They also find these mutant animals have a largely normal basic motor function. Given both the disease association of Neurexins and their broad functions in synaptic specification and maintenance, this work is necessary and of general interest. This manuscript would add to a growing body of literature on Neurexin function in neural circuits of central importance for neuropsychiatric disease. Given the context-specific functions of Nrxns, this work adds an important component. Major strengths are the impressive genetic tools, and the range of analyses employed. However, there are several weaknesses relating to both the technical approach (not using optically evoked DA transmission to study purely cell-autonomous effects) and under-sampling. Another issue (although less so) is the lack of attempts at any mechanistic understanding.

Here is a critique of the main claims and conclusions:

1. Mice with conditional Nrxn deletion have an unimpaired motor function – this is supported by the lack of motor phenotypes for motor learning (rotarod), although there are strong trends for the more difficult rotarod task, suggesting more challenging motor tasks might reveal deficits. The conclusion for these experiments should also talk about 'no changes to motor learning,' not just coordination.

2. Nrxn deletion mice have impaired locomotor responses to amphetamine – supported by the amphetamine result, which looks clear. It seems like a stretch to call this a 'DA-dependent behavior' though, it's really a pharmacological alteration. Do the authors want to speculate on why they need to use a pharmacological means to elicit a behavioral abnormality? Also, it's unclear whether sucrose preference can really probe motivational function in these mice, and whether enough has been done to say responses to natural rewards are altered.

3. Nrxn deletion mice have altered DA neurotransmission – the strongest supporting data for this is found in the FSCV data, as the ultrastructure (which is quite nice) and the DAT immunohistochemistry are better aimed at finding an underlying mechanism that leads to altered neurotransmission. The FSCV data are significantly weakened by the mixed source of DA release when using extracellular stimulation in acute slices – a significant portion (~50%) of this DA signal can be via ChIN-driven DA release (eg. Figure 1C in Brimblecombe et al., eNeuro). Here, cholinergic neurons should be normal, which may lead to an underestimation of the phenotype, important given that peak overflow seems unchanged, and the phenotype is mainly seen in kinetics and from multiple pulses (where later pulses recruit more DA-axon DA release). It would be good to unpack what the authors think is happening with the changes in short-term plasticity – does this reflect a presynaptic release phenotype? Re: altered DA neurotransmission, I think the authors should further emphasize that this is largely a vSTR phenotype – this seems to get lost but is the most accurate interpretation of the data.

4. Deletion mice have reduced DA reuptake following activity-dependent release – this is the most convincing aspect of the alterations in neurotransmission. However, is it possible that the short-term plasticity phenotype relates to an interaction of this system with the cholinergic system?

5. Nrxn deletion mice have increased GABA co-release from DA terminals – this claim is supported by the increased optogenetically-elicited GABA'ergic responses. Given this is a viral expression, these data would be stronger if the authors confirm equal ChR expression. Further, there is under-sampling here with significant outliers.

6. Increased GABA co-release comes from an increase in GABA uptake – this claim depends on the reliability of this assay. Can the authors demonstrate that this assay uses similar principles as in vivo? This would enhance these findings.

7. Nrxns do not act as drivers of axon terminal or synapse formation – while this would be in keeping with the bulk of existing literature, this claim depends on when the Nrxn recombination is happening in relation to dopamine neuron axonal and synapse development. These are important details for LOF analysis.

– A lot of the statistical reporting for Figure is unclear in the text – in the 2way ANOVAs, are these p-values reporting a main effect? Interaction?

– Why is there no learning for the 4-40 rotarod? This result conflicts with many other publications.

– Also for the FSCV, I cannot understand how the short-term paired pulse is calculated. With this formula, it seems that PP depression should be a negative number?

– What underlies this short-term plasticity change? Is this a presynaptic change or a change in the manner that the cholinergic system interacts with DA terminals?

– It would be good to look at the FSCV with optically-evoked DA, as this will more cleanly test the cell-autonomous role of Nrxns in DA neurons.

– Alternatively, the authors could examine striatal DA levels in the presence of full cholinergic blockade.

– Related to the above, can the authors test whether the short-term plasticity of DA release phenotype reflects release probability changes in DA neurons or altered interactions with local cholinergic signaling? (ie. could Nrxn lead to mislocalization of nAChRs that are important in shaping striatal DA release in response to >1 stimuli?)

– The time course of Nrx deletion in DAT-Cre should be described and related to the time course of DA neuron innervation of the striatum. If the deletion is after initial striatal innervation by DA terminals, the conclusion that Nrxns have no role in this function should be softened.

– Given the split in WT data, the experiment in dSTR is likely underpowered to detect a reliable change – Fig5E dSTR data should be properly powered.

– For the GABA uptake assay, can the authors demonstrate that this assay uses similar principles as in vivo – can uptake be blocked by known membrane transporters or VMAT? This would enhance these findings.

– Authors should carefully go through the manuscript and make clear the distinctions between the effects seen in the dSTR and vSTR – sometimes they make general comments (striatal DA transmission) that don't reflect this regional distinction.

– Injection density of ChR should be quantified and shown to be roughly equivalent between GTs for opto GABA results.

[Editors’ note: further revisions were suggested prior to acceptance, as described below.]

Thank you for resubmitting your work entitled "Conditional deletion of neurexins dysregulates neurotransmission from dopamine neurons" for further consideration by *eLife*. Your revised article has been evaluated by Lu Chen (Senior Editor) and a Reviewing Editor.

The manuscript has been improved but there are some remaining issues that need to be addressed, as outlined below:

The reviewers have made detailed suggestions on how to improve the manuscript further. Please revise accordingly, and provide point-by-point responses to the reviewers with your revised manuscript.

*Reviewer #1 (Recommendations for the authors):*

In the revised version of the manuscript, the authors have made changes that address several of the reviewers' main comments – in particular the addition of the DHBE experiments for the FCV analysis and adding "n" to the ChR2 oIPSC experiments. They have also made some text revisions and clarifications. These changes have improved the manuscript but some of the same weaknesses pointed out in the original review persist. These include the general lack of mechanistic insight and challenges with generating an integrated conceptual framework for what NRXNs do in DA neurons. Also the relatively subtle (GABA uptake in cultures) and variable (region-specific oIPSC changes) results remain for some of the assays. There is value, however, in convincingly showing what NRXNs don't do in DA neurons, e.g. regulate axonal guidance, synapse formation, glutamatergic transmission, axon terminal ultrastructure, etc. Another strength is the comprehensive set of analyses performed. Also, there is novelty as this is the first study to describe the effects of NRXN loss in DA neurons. Overall I don't have major concerns about publishing this study; however, the impact may be limited in scope.

There are a few remaining points that should be addressed prior to publication:

1) In the abstract – the authors state "…a large subset of non-synaptic release sites and a smaller subset of synaptic terminals from which glutamate or GABA are released". DA can also be released at synaptic sites (albeit a small number) so perhaps this could be revised.

2) In the new FACS/RNA-seq experiment, it is unclear how SNc and VTA neurons were separated. This is not described in the methods. The authors should validate accurate separation of these populations by showing differential expression of a region-specific marker. It is also difficult to interpret raw read counts without normalizing to some control (e.g. a housekeeping gene that is similarly expressed in SNc and VTA neurons).

3) For the rotarod assay, the authors compare performance on the first versus last session. A potentially more robust metric that is commonly used is to measure the slope of performance for each mouse.

4) On page 9 – the behavior results do not really show a change in "DA neurotransmission" – perhaps the authors could revise this to specify that they observed an "altered response to psychostimulant challenge" (or something similar).

5) In the last sentence on page 13 – the authors could specify that only DA terminals were evaluated (not GABA-ergic or glutamatergic)

6) For the measurements of reuptake rate – was this measured from matched peak evoked transients? This is important as greater DA release is associated with faster reuptake.

7) The new results in Figure 5E are a bit difficult to interpret. The example traces show a much smaller current in the mutants, which is also reflected in the mean (although not significant). However, there are a few very large responders in the WT condition that drive up the average. If those were not considered, then the KO average would actually be higher. It's difficult to conclude from this data that there is definitely not an effect in dSTR.

8) In Figure 7, how were the SNc and VTA neurons separated or identified in the cultures? Were independent cultures prepared from these two regions? There does not appear to be a methods section for the cell culture and GABA uptake experiments.

9) The inclusion of Sup. Figure 1 is helpful to understand the breeding scheme. However, showing DAT-Cre and the floxed NRXN isoforms on the same "allele" is not accurate. These are on different chromosomes and would be inherited independently.

*Reviewer #3 (Recommendations for the authors):*

Ducrot et al. have described synaptic, morphological and behavioral phenotypes in mice lacking all Neurexins within midbrain dopamine neurons. The disease-related importance of these proteins and the dopaminergic cell types being examined make these studies broadly interesting to the field. The revised manuscript is improved from the original submission, particularly regarding the evoked dopamine release. Strengths of the manuscript include a clear demonstration of reductions in evoked striatal dopamine release as well as an increase in the evoked GABAergic synaptic transmission from vSTR-targeting DA neurons. Weaknesses of the manuscript include minimal understanding of the relationship of these changes to behavior, confusing mechanistic insights into the increase in GABAergic synaptic transmission and no documentation of time course of the loss-of-function.

1. Figure 7, the mechanistic understanding of the enhanced GABA release – a central finding in the current framing – is confusing and raises questions about the meaning of this assay. Optically-evoked DA release is increased in the VStr but unchanged (in fact, strongly trending towards a decrease) in the dSTR. However, Figure 7 shows that the SNc axons (a) have much higher amounts of surface GABA and (b) take up a similar amount of GABA as compared to VTA axons. Given this, it is hard to see how this supports this potential hypothesis for the increase in GABA release from DA neurons in vSTR.

2. is there actual data supporting the idea that Nrxn is being deleted before synapse formation? When are the Nrxn transcripts or protein no longer detected? Claiming that the Cre turns on when the DAT gene turns on makes many untested assumptions.

---

## [Author Response]

[Editors’ note: the authors resubmitted a revised version of the paper for consideration. What follows is the authors’ response to the first round of review.]

Comments to the Authors:We are sorry to say that, after consultation with the reviewers, we have decided that this work in its current form will not be considered further for publication by eLife.The reviewers acknowledge the overall significance of the work, though they nevertheless all have several important comments. In particular, they feel that at present the study is somewhat underpowered because of the low N numbers in several experiments and that it would be important to have more rigorous controls. We discussed these points among the reviewers, and we take these points seriously and have the opinion that they need to be carefully experimentally addressed to ensure rigor.However, if you can fully address the reviewer's concerns, we would be happy to encourage you to submit a revised manuscript as a new submission. We hope that you will find the reviewers' comments to be constructive and if you have any questions about the reviews please let us know.Reviewer #1 (Recommendations for the authors):By generating a mouse model in which neurexins (Nrxns) are selectively deleted from dopamine (DA) neurons, Ducrot et al. explore the role of these presynaptic adhesion proteins in neurotransmission from DA neurons. They show that mice in which Nrxn 1, 2, and 3 are deleted from DA neurons display a significant reduction in amphetamine-induced locomotion, but no alterations in basal movement, motor coordination/learning on the rotarod, and cocaine-induced activity. Immunohistochemistry revealed that Nrxns KO mice show an increase in VMAT2 and a reduction in DAT-positive (but not intensity) in ventral striatum (vStr), but not dorsal striatum (dStr). Using fast-scan cyclic voltammetry, the authors reveal intact DA release but a reduced rate of DA uptake in vStr but not dStr. Moreover, the authors find that GABA co-release from DA neuron is increased in vStr of Nrxns KO mice, while glutamate transmission is not affected.Major Strengths:This study combines an impressive array of experimental methods to characterize these triple conditional knockout mice, including behavioral assays for basic locomotor function, immunohistochemistry of presynaptic DA markers, electron microscopy of DA synapse ultrastructure, fast-scan cyclic voltammetry of DA release, and whole-cell electrophysiology to monitor GABA and glutamate co-release. In most cases, analyses are applied to ventral and dorsal striatal regions for a comprehensive characterization of synaptic function from mesolimbic and nigrostriatal dopaminergic neurons.Major Weaknesses:This study is mainly concerned with providing a list of differences between control and knockout mice. While the number of assays used is impressive, the study does not deeply investigate any of the reported phenotypes, some of which are quite small in magnitude and suffer from low numbers of experimental observations. It is for instance unclear how the immunohistochemical findings relate to the behavior or voltammetry/electrophysiological data, why deficits are limited to the ventral striatum, or what the relative contributions to neurexins 1, 2, or 3 to the many phenotypes reported here are. Overall, this descriptive study is limited in the conclusions it can make, falling short of markedly increasing our understanding of the effects of neurexin 1, 2, or 3 on synaptic function.

We thank the reviewer for her/his globally positive opinion on our work. We agree that we have not figured out all of the links between the different observations but considering the complex roles of neurexins in cell-cell communication, such an ambitious goal could take years. We believe that the present set of results reinforce the concept that one of the roles of neurexins is to regulate the neurotransmitter repertoire of neurons and we are confident the results will trigger more work in the field.

Specific points:1. It is not clear which neurexins midbrain dopamine neurons express, and whether differences exist between VTA and SNc neurons that may explain the lack of phenotype in SNc neurons? The authors first need to confirm which Neurexin(s) is expressed in DA neurons and that they are effectively lost from DA neurons in the knockout.

We modified figure 1 to include results from a RNASeq experiment that we performed from FACS-sorted dopamine neurons obtained from TH-GFP mice. Panel A now shows the levels of neurexin 1-2-3 in VTA and SNc dopamine neurons. We added a complete description of the RNA sequencing experiment in the STAR methods (bottom of page 37).

2. It is not clear how immunohistochemical changes in DAT or VMAT2 'surface' relate to DA neurotransmission, especially when more classical measures of protein abundance like intensity are not changed. For example, the authors suggest the decrease in DAT 'surface' may be related to the slower time constant of DA uptake observed with fast-scan cyclic voltammetry, but slowing of DA uptake is not accompanied by changes in DAT 'surface' in the dorsal striatum. Similarly, it is unclear why an increase in VMAT2 'surface' does not translate into greater DA release. Lastly, since the authors report a GABA co-release phenotype, the authors should complement their immunohistochemical analyses with protein targets related to GABA co-release. Together, they will help make sense of the phenotypes observed by suggesting mechanisms.

A change in DAT or VMAT2 signal surface without a change in signal intensity could for example represent a change in the density of dopamine neuron terminals, as previously reported in one of our publications (Giguere et al., 2019, PMID: 31449520). In any case, with the new cyclic voltammetry data (obtained under nicotinic blockade) showing that the rate of dopamine reuptake is not significantly changed in the KO mice, we have now revised the discussion of the manuscript and propose a new interpretation of the slowed kinetics of dopamine overflow under normal saline conditions (pages 23-24).

We appreciate the suggestion to examine membrane GABA transporter levels in the KO mice, However, considering the very high density of this protein in the striatum, we considered that unless extensive additional ultrastructural experiments were performed, it would be very difficult to selectively quantity the GAT-1 signal coming from dopamine neuron terminals. We hope the reviewer will understand that further experiments outside the scope of the present manuscript will be necessary to solve the mystery of the increase in GABA release by dopamine neurons in the neurexin KO mice.

3. The authors report a potentiation of GABA co-release and a slowing down of IPSC decay kinetics in the vStr of Nrxn KO mice compared to WT, while dStr responses are unaffected. However, both data sets suffer from low Ns with a handful of outliers that may significantly skew population averages and confound interpretation. For example, Figures 5C and D each contain 3 data points that are significantly larger than all others. Are they from the same mouse, which may have had better virally-mediated ChR2 expression than the rest? The same is true for Figure 5E in dStr. Would the findings in vStr and dStr stand without these outliers? Given the observed variability of IPSC response magnitude when ChR2 is expressed virally, any conclusion regarding GABA co-release would need to be bolstered with larger Ns across more animals.

As requested by the reviewers, we bolstered our conclusion by preparing more animals to increase the Ns. After injecting a new cohort of DAT::NrxnsKO and DAT::NrxnsWT animals with the AAV5-DIO-ChR2-EYFP, we performed a new set of patch-clamp recording to increase significantly the Ns for electrophysiology experiments. The revised statistical analyses confirm the increase of GABA co-release by DA neurons after optogenetic stimulation in the vSTR but not in the dSTR. Figure 5 has been revised accordingly.

4. In Figure 7, the authors conclude that GABA uptake is much higher in Nrxns KO mice compared to WT mice. However, this result arises from a main effect of genotype following a 2-Way ANOVA, which pools together SNC and VTA data. Is a statistically significant increase still seen when simply comparing KO vs WT mice in SNc cultures, and separately KO vs. Wt mice in VTA cultures? The differences appear minor and are unlikely to explain the IPSC phenotype observed vStr but not dStr in Figure 5. Wouldn't an increase in GABA uptake speed up the decay time constant? How can the authors rule out the increase in VMAT2 'surface' underlying the latter?

We agree that the increase in GABA uptake in the KO cells is modest and significance comes out as an overall genotype effect. We are not able to separate the analysis of SNc and VTA results because this would not be according to best practices. Considering that a complete pharmacological blockade of GAT-1 is required to slow down evoked IPSC kinetics (PMID: 15987761), a small increase in GAT levels would not be expected to accelerate the time constant of IPSC decay because other factors such as diffusion also play key roles in this process. We have clarified our interpretation of the changes in oIPSC decay time (but not of decay time constant) on page 18. We agree that the increase in VMAT2 signal we detected could also play a role in the increase in GABA release we observed. This is mentioned in the discussion on page 26 and in the new figure 8 schematic diagram on last page of the discussion (page 27).

5. DA Voltammetry is performed using electrical stimulation, which preferentially stimulates cholinergic interneurons and drives DA release indirectly via nicotinic acetylcholine receptors. This mechanism complicates analyses of DA release probability. The authors should repeat these analyses in the presence of DHBE or other nAChR antagonist to evoke DA release from DA axons directly. Only then can the release probability of DA from DA axons be directly evaluated.

We have considered the reviewer’s comment regarding the role of cholinergic neurons in DA release. We repeated all of our voltammetry recordings in the presence of D_β_HE. These results were helpful because they revealed that direct DA release detected under such conditions was reduced in the KO mice. This also allowed us to reveal that the direct rate of DA reuptake is unchanged in the KO mice. Our findings fit nicely with the results of a similar study performed in serotonin neurons and just recently published in *eLife* (PMID: 36695811). The new results are presented in the revised figure 4.

1. Figure 2: In order to better appreciate the differences between WT and cKO in VMAT2 and Dat expression, the authors should provide more informative and representative images, possibly with higher contrast. For example, by looking at the images shown in Figure 2B, it looks like VMAT2 is fainter in dStr of WT mice than in cKO. Also in Figure 2D, Dat signal seems much lower in cKO. In addition, there is a typo in the y-axis of Figure 2I as I am assuming it should read "DAT signal surface".

As requested by the reviewers, we provided images with higher contrast for VMAT2 and DAT signals in the revised figure 2. We also corrected the typo in the y-axis of Figure 2I.

2. Methodological details explaining the immunohistochemistry is quantified are missing.

We added more details about the IHC signal quantification in the *STAR methods* section (bottom of page 38).

3. Page 10, line5, the authors say "We focused on terminals in the vStr because this area contains DA neuron release sites for both glutamate and GABA in addition to those for DA". This statement is misleading as it suggests that both glutamate and GABA from DA neurons are only released in vStr.

We changed the sentence to the following: “We focused on terminals in the vSTR, where we observed significant changes in VMAT2 and DAT, and which is the most characterized brain region showing DA neuron-mediated glutamate and GABA co-transmission (Stuber et al., 2010; Berube-Carriere et al., 2012).” (page 11 of the revised manuscript).

4. There is a typo on page 22, line 14: "regulators if glutamate co-release".

This has been corrected.

5. Please show traces of paired-pulse depression in Figure 4 or in supplemental materials.

As requested by the reviewers, we provided example FSCV traces for the paired-pulses experiments. This can be found in supplementary figure S3, panels K to P. But just to be clear, in response to paired pulses, the individual responses cannot be resolved.

6. The increase in sIPSC frequency in vSTr of Nrxns cKO mice is not supported by an appropriate hypothesis. Although in the Discussion the authors state that "our observation of an increase in sIPSC frequency in the vSTR of KO mice could potentially result from alterations in DA neuron projections to non-striatal regions", no experiments are performed to support this, and the findings have little to no bearings on the rest of the study. I suggest removing these data.

As requested, we have now removed the previous supplementary figure 6 from the manuscript.

Reviewer #2 (Recommendations for the authors):The goal of this study was to determine how deletion of the genes encoding the neurexin family of trans-synaptic adhesion molecules (Nrxns), impacts neurotransmitter release from dopamine neurons. The authors used multiple approaches to provide a comprehensive overview of the consequences of Nrxn loss on the output of the dopaminergic system. Specifically, they show that while overall axonal architecture and dopamine release are unaffected by Nrxn loss, there are alterations in DAT function that impair dopamine re-uptake in the striatum. This leads to altered locomotor activation in response to amphetamine without affecting baseline motor function.In addition to measuring dopamine, this study assesses GABA and glutamate release, which are co-transmitters in dopamine neurons. The authors make the interesting observation that GABA release from dopamine neurons is enhanced by loss of Nrxns, while glutamate release is unaltered. This potentiation of GABA release is notably region-specific and only occurs in the ventral but not dorsal striatum. The study goes on to show that GABA uptake from cultured Nrxn knock-out dopamine neurons is enhanced, suggesting an intriguing mechanism by which GABA signaling may be potentiated. Overall, this study provides the first look at how disruption of Nrxns affects dopamine neuron output, as these molecules have until now been studied on glutamatergic or GABA-ergic neurons, with complex effects on synaptic development and function.The manuscript is well-written and the data generally support the conclusions (although see limitations below). The discussion of the data is balanced and the discussion highlights areas that warrant further investigation. The data is of good quality and the assays used represent the standard in the field and cover an array of dopamine neuron properties (e.g cellular morphology, subcellular ultrastructure, dopamine release & re-uptake, GABA, and glutamate release, motor behavior, and drug-induced behavior). The use of the GABA uptake assay in dopamine neurons is particularly innovative and compelling, especially as dopamine neurons are challenging to culture.There are a few limitations of the study. Specifically, it is unclear whether DAT-Cre positive mice were used as controls or whether the controls were DAT-Cre negative. The presence of Cre reduces the expression of endogenous DAT in DAT-Cre mice, which alters dopamine release and behavior. Cre expression, therefore, needs to be present in all groups for an accurate comparison. If Cre+ controls were used, it would be helpful for the authors to describe their breeding scheme. If the Nrxn floxed/DAT-Cre+ mice were maintained as a separate line from Nrxn WT/DAT-Cre+ mice, this should be noted.For the recordings of GABA and glutamate release within the striatum, unidentified striatal neurons (SPNs) were recorded. The two main types of SPNs can differ significantly in their intrinsic and synaptic properties; therefore, a more rigorous approach would be to identify and record these neurons separately. In addition, some of the physiology experiments use a relatively small number of neurons (<10). Given that the ChR2 is delivered virally and there could be significant differences in viral transduction between mice, a larger sample size (or control for viral expression) would be needed to strengthen the conclusions. The FSCV experiments use electrical stimulation but the effects of acetylcholine release on dopamine transmission have not been controlled for. It would be helpful to repeat these experiments using ChR2 stimulation or with cholinergic blockers to exclude any contribution of altered cholinergic signaling to the observed phenotypes. Finally, while this study provides an important characterization of the consequences of Nrxn loss, it does not delve into the mechanistic basis of these changes, which would be an interesting area for future investigation. Specifically, it would be interesting to know whether GABA membrane transporter expression (Gat) is altered in Nrxn KO dopamine neurons and how Nrxns interact with DAT.Specific points:1) It is not clear whether the control mice in this study were positive for DAT-Cre. i.e. were the controls WT for all Nrxns and DAT-Cre+? This is important as DAT expression and function are reduced in DAT-Cre+ mice, which affects dopamine release, reuptake, and basic locomotor behavior (e.g. see PMIDs: 16865686 and 33979604). This could be a confound if the Nrxn KO mice were DAT-Cre positive but the controls were DAT-Cre negative. If Cre+ controls were used, it would be helpful for the authors to describe their breeding scheme. If the Nrxn floxed/DAT-Cre+ mice were maintained as a separate line from Nrxn WT/DAT-Cre+ mice, this should be noted. It seems that it would be very challenging to generate littermate mice that were homozygous floxed or homozygous WT for all three Nrxn genes and also DAT-Cre positive.

We have now added a full description of the breeding scheme to explain the generation of DAT::NrxnsKO by crossing Nrxn 123^flox^ mice with DAT^IRES-CRE^ mouse line – The breeding scheme can be found in supplemental figure S1.

2) The authors suggest that the sucrose preference task measures motivation. My understanding is that this test is a measure of hedonia/anhedonia rather than purely "motivation". The authors may want to revise the text here or consider a more classical test of motivation, e.g. lever pressing progressive-ratio test.

We revised the text and now describe this test as a measure of hedonia/anhedonia rather than motivation (page 7).

3) For many of the graphs, the authors have only shown error bars in one direction. It would be more rigorous to show the error bars in both directions. This allows easy assessment of whether the error bars overlap (typically denoting a lack of a significant difference between groups).

As requested, we modified all bar graphs to show the error bars in both directions. We only kept unidirectional bars for some of the behavioral performance graphs in figure 1, which became less legible with two-sided bard.

4) The authors should provide example FSCV traces for the paired-pulse experiments in Figure 4. In addition, the authors should control for any potential contribution of cholinergic alterations on dopamine release and reuptake. This could be done by including cholinergic blockers or by using ChR2 to evoke dopamine release.

As requested by the reviewers, we now provide example FSCV traces for the paired-pulses experiments. This is presented in supplemental figure 3. But just to be clear, in response to paired pulses, the individual responses cannot be resolved. We have also added a new series of experiments performed using a nicotinic blocker. This is presented in the revised figure 4.

5) The physiology results suggest an interesting increase in GABA but not glutamate release from dopamine neurons. These conclusions are based on a relatively small sample size for some experiments (<10 neurons) and there could be considerable variability in viral transduction across mice. It would strengthen the conclusions if more "n" could be added to these experiments.

As requested by the reviewers, we bolstered our conclusion by adding more animals. We injected a new cohort of DAT::NrxnsKO and DAT::NrxnsWT animals with the AAV5-DIO-ChR2-EYFP and performed a new set of patchclamp recording to increase significantly the Ns for electrophysiology experiments. The revised statistical analyses confirm our original conclusion of increased GABA co-release by DA neurons after optogenetic stimulation in the vSTR but not in the dSTR.

6) For Figure 5, it would be helpful to show a higher magnification image of EYFP expression in the midbrain – to show that it is exclusively expressed in TH positive dopamine neurons and not neighboring GABA or glutamatergic neurons.

As requested by the reviewers, in the revised figure 5, we provided a new set of images with a higher magnification, better showing the EYFP expression in the midbrain. We also performed quantifications of the colocalization between the EYFP and TH or DAT signals, which is provided in the new supplemental figure 5. Colocalization was close to 100%.

7) The GABA uptake experiments in Figure 7 are an innovative approach to assessing the potential mechanisms for increased GABA-ergic transmission from DA neurons. The authors mention that DA neurons were co-cultured with SPNs. How is GABA uptake affected in SPNs? i.e. We would assume that GABA uptake should only be enhanced in DA neurons from DAT-Nrxn KO, not the co-cultured SPNs. This would provide strong evidence that changes in the DA neurons themselves are responsible for enhanced oIPSCs.

In these experiments, we only obtained images from fields that contained dopamine neurons. As such, there were only very few neurons that were non-dopaminergic in these image sets. As such, it is not possible for us to conclude on the GABA uptake by SPNs. As requested, we modified the title of figure 7.

8) How were the statistical comparisons done for Figure 7? From the graph, it is unclear which groups are statistically different. This is important as the effect size is fairly small.

This was done using a 2-way ANOVA. The genotype effect is significant, but not the difference between the two regions is not. This has been clarified in the figure 7 legend.

9) Given the potential mechanism shown in Figure 7, it would be very interesting to know whether the expression of Gat1 is altered in DAT-Nrxn KO neurons, although it is understood that this may be beyond the scope of this study.

As per our response to reviewer 1, we appreciate the suggestion to examine membrane GABA transporter levels in the KO mice. However, considering the very high density of this protein in the striatum, we considered that unless extensive additional quantitative ultrastructural experiments were performed, it would be very difficult to selectively quantity the GAT-1 signal coming from dopamine neuron terminals. We hope the reviewer will understand that further experiments outside the scope of the present manuscript will be necessary to solve the mystery of the increase in GABA release by dopamine neurons in the neurexin KO mice.

Reviewer #3 (Recommendations for the authors):Overall, this is an interesting manuscript that uses established techniques to describe the impact of pan-Neurexin deletion on the cell-autonomous function of dopamine neurons in the mouse brain. Using cell type-specific conditional gene deletion, they find that Neurexins play a role in maintaining aspects of DA neurotransmission, particularly within the ventral striatum. They also find these mutant animals have a largely normal basic motor function. Given both the disease association of Neurexins and their broad functions in synaptic specification and maintenance, this work is necessary and of general interest. This manuscript would add to a growing body of literature on Neurexin function in neural circuits of central importance for neuropsychiatric disease. Given the context-specific functions of Nrxns, this work adds an important component. Major strengths are the impressive genetic tools, and the range of analyses employed. However, there are several weaknesses relating to both the technical approach (not using optically evoked DA transmission to study purely cell-autonomous effects) and under-sampling. Another issue (although less so) is the lack of attempts at any mechanistic understanding.Here is a critique of the main claims and conclusions:1. Mice with conditional Nrxn deletion have an unimpaired motor function – this is supported by the lack of motor phenotypes for motor learning (rotarod), although there are strong trends for the more difficult rotarod task, suggesting more challenging motor tasks might reveal deficits. The conclusion for these experiments should also talk about 'no changes to motor learning,' not just coordination.

As requested by the reviewer, we revised the text and now refer to motor learning (page 6).

2. Nrxn deletion mice have impaired locomotor responses to amphetamine – supported by the amphetamine result, which looks clear. It seems like a stretch to call this a 'DA-dependent behavior' though, it's really a pharmacological alteration. Do the authors want to speculate on why they need to use a pharmacological means to elicit a behavioral abnormality? Also, it's unclear whether sucrose preference can really probe motivational function in these mice, and whether enough has been done to say responses to natural rewards are altered.

We understand the reviewer’s questioning. However, because our investigation targeted dopamine neurons and because amphetamine directly targets dopamine transporters and causes extracellular dopamine elevation, it appeared to us as natural to use this approach to evaluate the state of the dopamine system in these mice. This is an approach commonly used in the field (as per the work referenced on page 5). We nonetheless toned tone down the statement regarding DA-dependent behaviors (page 7). We also clarified that the sucrose preference test was used to evaluate hedonia and not motivation (page 7).

3. Nrxn deletion mice have altered DA neurotransmission – the strongest supporting data for this is found in the FSCV data, as the ultrastructure (which is quite nice) and the DAT immunohistochemistry are better aimed at finding an underlying mechanism that leads to altered neurotransmission. The FSCV data are significantly weakened by the mixed source of DA release when using extracellular stimulation in acute slices – a significant portion (~50%) of this DA signal can be via ChIN-driven DA release (eg. Figure 1C in Brimblecombe et al., eNeuro). Here, cholinergic neurons should be normal, which may lead to an underestimation of the phenotype, important given that peak overflow seems unchanged, and the phenotype is mainly seen in kinetics and from multiple pulses (where later pulses recruit more DA-axon DA release). It would be good to unpack what the authors think is happening with the changes in short-term plasticity – does this reflect a presynaptic release phenotype? Re: altered DA neurotransmission, I think the authors should further emphasize that this is largely a vSTR phenotype – this seems to get lost but is the most accurate interpretation of the data.

We have considered the reviewer’s comment regarding the role of cholinergic neurons in DA release. We repeated all voltammetry recordings in the presence of DβHE. This revealed that direct DA release detected under such conditions was reduced in the KO mice. The results also allowed us to reveal that the direct rate of DA reuptake is unchanged in the KO mice. Our findings fit nicely with the results of a similar study performed in serotonin neurons and just recently published in *eLife* (PMID: 36695811). The new results are presented in the revised figure 4.

4. Deletion mice have reduced DA reuptake following activity-dependent release – this is the most convincing aspect of the alterations in neurotransmission. However, is it possible that the short-term plasticity phenotype relates to an interaction of this system with the cholinergic system?

We agree that changes in DA overflow and its plasticity in the KO mice could depend in part on changes in the interaction between the DA terminals and the striatal cholinergic system. In the revised discussion, we now refer to this possibility (pages 23-24).

5. Nrxn deletion mice have increased GABA co-release from DA terminals – this claim is supported by the increased optogenetically-elicited GABA'ergic responses. Given this is a viral expression, these data would be stronger if the authors confirm equal ChR expression. Further, there is under-sampling here with significant outliers.

We agree that with viral expression of ChR2, there could be some variability in ChR2 expression. To account for the additional variance related to this, we performed additional experiments and increased the number of observations in each group. The new data confirm our original findings. The results were added to the revised figure 5. We have also added new illustrations of the ChR2-YFP expression in figure 5 and found that it is comparable in all mice examined and well expressed in DA neuron axons (Supplemental figure 5).

6. Increased GABA co-release comes from an increase in GABA uptake – this claim depends on the reliability of this assay. Can the authors demonstrate that this assay uses similar principles as in vivo? This would enhance these findings.

We have not compared the characteristics of this in vitro GABA uptake assay to the properties of GABA uptake in vivo. This would require a lot of additional work. The validity of the in vitro assay is at least validated by the fact that the levels of GABA immunoreactivity in dopamine neurons in these experiments was robustly increased by incubating the neurons in GABA, as shown by figure 7, panel B.

7. Nrxns do not act as drivers of axon terminal or synapse formation – while this would be in keeping with the bulk of existing literature, this claim depends on when the Nrxn recombination is happening in relation to dopamine neuron axonal and synapse development. These are important details for LOF analysis.

We now state in the revised manuscript that “Considering that the DAT gene is turned on at around embryonic days 14-15, the gene deletion is expected to have happened at early stages of the establishment of DA neuron connectivity. More extensive changes in the basic connectivity of DA neurons could perhaps have been detected with an earlier KO.” (page 22).

– A lot of the statistical reporting for Figure is unclear in the text – in the 2way ANOVAs, are these p-values reporting a main effect? Interaction?

We have now clarified the statistical reporting.

– Why is there no learning for the 4-40 rotarod? This result conflicts with many other publications.

We found robust learning on the rotarod task when the speed was increased from 4-40 over 10 min (Figure 1C-D-E-F).

It is only when the speed increased from 4-40 over a much shorter 2 min period that the mice failed to learn (Supplementary Figure 2A-B-C-D). We thus conclude that with this faster version, the mice we used found the task too challenging and failed to improve. This more demanding version of the rotarod is not often used. In a previous publication, mice were able to learn a similar demanding version of the rotarod (8-80 rpm), but over a 5 min period (PMID 24995986). Since our apparatus could not reach such high rpm values, we shortened the duration of the task instead.

– Also for the FSCV, I cannot understand how the short-term paired pulse is calculated. With this formula, it seems that PP depression should be a negative number?

When a single pulse (P1) is given, the dopamine overflow detected is quite large, as shown by our recordings. When such a single stimulation is replaced by two closely spaced pulses (P2), the second of these two pulses only leads to very small additional dopamine release because most of the releasable pool was released on the first of these two pulses. This reflects strong paired-pulse depression. As described in the methods section, we calculate the paired pulse ratio by subtracting P1 from the P2 double stimulation (P2-P1) and we divide this by the P1 value. In normal ACSF, this typically gives a fraction between 0.1 and 0.2, reflecting the fact that the second pulse only released 1020% of the amount of dopamine released by the first pulse. This is now explained in the revised STAR methods section (page 35).

– What underlies this short-term plasticity change? Is this a presynaptic change or a change in the manner that the cholinergic system interacts with DA terminals?

This is not known for sure. But much of this strong paired-pulse depression is thought to depend on depletion of vesicular pools in DA terminals. We did not explore this further here as this is outside the scope of the present manuscript.

– It would be good to look at the FSCV with optically-evoked DA, as this will more cleanly test the cell-autonomous role of Nrxns in DA neurons.

This is a good suggestion. However, instead of this, we performed a new series of FSCV recordings in the presence of a nicotinic receptor antagonist, thus isolating direct DA release. These results are presented in the revised figure 4.

– Alternatively, the authors could examine striatal DA levels in the presence of full cholinergic blockade.

Yes. Please see our response to the previous point.

– Related to the above, can the authors test whether the short-term plasticity of DA release phenotype reflects release probability changes in DA neurons or altered interactions with local cholinergic signaling? (ie. could Nrxn lead to mislocalization of nAChRs that are important in shaping striatal DA release in response to >1 stimuli?)

This is an intriguing suggestion and our observation that the slower recovery kinetics of DA overflow seen in the KO mice is not observed under nicotinic blockade does argue for an indirect effect through the cholinergic system. This is now discussed in the revised discussion (pages 23-24).

– The time course of Nrx deletion in DAT-Cre should be described and related to the time course of DA neuron innervation of the striatum. If the deletion is after initial striatal innervation by DA terminals, the conclusion that Nrxns have no role in this function should be softened.

We agree that this is an issue that would be worth discussion more. However, considering the already long discussion, we have only added a short comment to specify when the DAT promoter is turned on and that a more extensive phenotype could perhaps have been observed with an earlier KO (page 22).

– Given the split in WT data, the experiment in dSTR is likely underpowered to detect a reliable change – Fig5E dSTR data should be properly powered.

As requested, we have increased the number of experiments. Our main conclusions are confirmed.

– For the GABA uptake assay, can the authors demonstrate that this assay uses similar principles as in vivo – can uptake be blocked by known membrane transporters or VMAT? This would enhance these findings.

We have not compared the characteristics of this in vitro GABA uptake assay to the properties of GABA uptake in vivo. This would require a lot of additional work. We have unfortunately also not tried to block the uptake using GABA transporter blockers. The validity of the in vitro assay is at least validated by the fact that the levels of GABA immunoreactivity in dopamine neurons in these experiments was robustly increased by incubating the neurons in GABA, as shown by figure 7, panel B.

– Authors should carefully go through the manuscript and make clear the distinctions between the effects seen in the dSTR and vSTR – sometimes they make general comments (striatal DA transmission) that don't reflect this regional distinction.

We went through the manuscript to make this more consistent.

– Injection density of ChR should be quantified and shown to be roughly equivalent between GTs for opto GABA results.

We agree that with viral expression of ChR2, there could be some variability in ChR2 expression. To account for the additional variance related to this, we performed additional experiments and increased the number of observations in each group. The new data confirm our original findings. The results were added to the revised figure 5. We have also added new illustrations of the ChR2-YFP expression in figure 5 and found that it is comparable in all mice examined and well expressed in DA neuron axons (Supplemental figure 5).

[Editors’ note: what follows is the authors’ response to the second round of review.]

The manuscript has been improved but there are some remaining issues that need to be addressed, as outlined below:The reviewers have made detailed suggestions on how to improve the manuscript further. Please revise accordingly, and provide point-by-point responses to the reviewers with your revised manuscript.Reviewer #1 (Recommendations for the authors):In the revised version of the manuscript, the authors have made changes that address several of the reviewers' main comments – in particular the addition of the DHBE experiments for the FCV analysis and adding "n" to the ChR2 oIPSC experiments. They have also made some text revisions and clarifications. These changes have improved the manuscript but some of the same weaknesses pointed out in the original review persist. These include the general lack of mechanistic insight and challenges with generating an integrated conceptual framework for what NRXNs do in DA neurons. Also the relatively subtle (GABA uptake in cultures) and variable (region-specific oIPSC changes) results remain for some of the assays. There is value, however, in convincingly showing what NRXNs don't do in DA neurons, e.g. regulate axonal guidance, synapse formation, glutamatergic transmission, axon terminal ultrastructure, etc. Another strength is the comprehensive set of analyses performed. Also, there is novelty as this is the first study to describe the effects of NRXN loss in DA neurons. Overall I don't have major concerns about publishing this study; however, the impact may be limited in scope.

We appreciate the reviewer’s careful review and the highlighting of the strength and novelty of our study. We also agree that the scope is limited to one family of synaptic adhesion molecules within the context of DA neurons across two brain regions. This current study provides novel evidence and directions that help to set the stage for future studies with a wider scope.

There are a few remaining points that should be addressed prior to publication:1) In the abstract – the authors state "…a large subset of non-synaptic release sites and a smaller subset of synaptic terminals from which glutamate or GABA are released". DA can also be released at synaptic sites (albeit a small number) so perhaps this could be revised.

We thank the reviewer for pointing this out. We have revised the abstract to include DA accordingly on page 2, line 34.

2) In the new FACS/RNA-seq experiment, it is unclear how SNc and VTA neurons were separated. This is not described in the methods. The authors should validate accurate separation of these populations by showing differential expression of a region-specific marker. It is also difficult to interpret raw read counts without normalizing to some control (e.g. a housekeeping gene that is similarly expressed in SNc and VTA neurons).

We have provided additional information in the methods section (page 39 of the revised manuscript) and added three additional genes (Sox6, Slc17a6, and Calbn1) in Figure 1A. This additional analysis of region-specific markers provides further validation of the region-specific dissections. Further, we have also provided clarification for presenting the data as FKPM (revised Methods section, page 39). In brief, the DESeq2 method is a method using raw read counts that are normalized. As per the relevant DESeq2 manual (https://chipster.csc.fi/manual/deseq2.html): “DESeq2 performs an internal normalization where geometric mean is calculated for each gene across all samples. The counts for a gene in each sample is then divided by this mean. The median of these ratios in a sample is the size factor for that sample. This procedure corrects for library size and RNA composition bias, which can arise for example when only a small number of genes are very highly expressed in one experiment condition but not in the other.”

3) For the rotarod assay, the authors compare performance on the first versus last session. A potentially more robust metric that is commonly used is to measure the slope of performance for each mouse.

Although this is not very common in the literature, we agree with the reviewer that comparing the slope of the change in the latency to fall might provide another way to compare the data across genotypes. We have done this and see no significant difference in the slopes of WT and cKO mice. This is now reported in the revised text (page 6).

4) On page 9 – the behavior results do not really show a change in "DA neurotransmission" – perhaps the authors could revise this to specify that they observed an "altered response to psychostimulant challenge" (or something similar).

We agree with the reviewer and have modified the sentence that now reads “The finding of enhanced behavioural response to amphetamine suggests that loss of Nrxns in DA neurons leads to some alteration of the functionality of the DA system and some DA-dependent behaviors” on Page 7, lines 161-163.

5) In the last sentence on page 13 – the authors could specify that only DA terminals were evaluated (not GABA-ergic or glutamatergic)

We appreciate the reviewer pointing this out and allowing us to clarify. We have made the requested change (bottom of page 11of the revised manuscript).

6) For the measurements of reuptake rate – was this measured from matched peak evoked transients? This is important as greater DA release is associated with faster reuptake.

We thank the reviewer for this opportunity to clarify. No, the measurements of the reuptake rate were not measured from matched peak transients. This would be an issue if we had simply measured decay half time. However, instead, we quantified the tau values, which are more resistant to changes in signal amplitude. To ensure that this is clear, we have now stated more clearly that tau values were used and why this is a good choice (revised Methods section, page 36).

7) The new results in Figure 5E are a bit difficult to interpret. The example traces show a much smaller current in the mutants, which is also reflected in the mean (although not significant). However, there are a few very large responders in the WT condition that drive up the average. If those were not considered, then the KO average would actually be higher. It's difficult to conclude from this data that there is definitely not an effect in dSTR.

We aim to collect and report all data in an unbiased manner. While we recognize that there are some variations among each neuron recorded, upon careful statistical analyses, these variations do not fall under the outlier criteria and do not justify for data removal. As such, we decided to keep the whole data set and accept that there is no statistically significant difference between the amplitude of oIPSCs recorded in the dorsal striatum.

8) In Figure 7, how were the SNc and VTA neurons separated or identified in the cultures? Were independent cultures prepared from these two regions? There does not appear to be a methods section for the cell culture and GABA uptake experiments.

We have now added a section in the methods to describe the cell culture method (page 39 of the revised manuscript). In addition, as per point 2 above, our new data on gene expression differences of those region-specific markers validate the reliability of our VTA and SNc dissections.

9) The inclusion of Sup. Figure 1 is helpful to understand the breeding scheme. However, showing DAT-Cre and the floxed NRXN isoforms on the same "allele" is not accurate. These are on different chromosomes and would be inherited independently.

We agree with the reviewer and have now modified the supplementary figure 1 (now renamed Figure 1—figure supplement 1) to remove this possible confusion.

Reviewer #3 (Recommendations for the authors):Ducrot et al. have described synaptic, morphological and behavioral phenotypes in mice lacking all Neurexins within midbrain dopamine neurons. The disease-related importance of these proteins and the dopaminergic cell types being examined make these studies broadly interesting to the field. The revised manuscript is improved from the original submission, particularly regarding the evoked dopamine release. Strengths of the manuscript include a clear demonstration of reductions in evoked striatal dopamine release as well as an increase in the evoked GABAergic synaptic transmission from vSTR-targeting DA neurons. Weaknesses of the manuscript include minimal understanding of the relationship of these changes to behavior, confusing mechanistic insights into the increase in GABAergic synaptic transmission and no documentation of time course of the loss-of-function.1. Figure 7, the mechanistic understanding of the enhanced GABA release – a central finding in the current framing – is confusing and raises questions about the meaning of this assay. Optically-evoked DA release is increased in the VStr but unchanged (in fact, strongly trending towards a decrease) in the dSTR. However, Figure 7 shows that the SNc axons (a) have much higher amounts of surface GABA and (b) take up a similar amount of GABA as compared to VTA axons. Given this, it is hard to see how this supports this potential hypothesis for the increase in GABA release from DA neurons in vSTR.

We agree with the reviewer that this could be confusing given there are very little or no other studies that investigated this question and that provide further mechanistic hypotheses. We have now added one possible explanation and suggested future work in the revised discussion (pages 24-25 of the revised manuscript).

2. is there actual data supporting the idea that Nrxn is being deleted before synapse formation? When are the Nrxn transcripts or protein no longer detected? Claiming that the Cre turns on when the DAT gene turns on makes many untested assumptions.

We agree with the reviewer that we do not know precisely when the Nrxns are deleted in our mice. We cannot reliably measure when the Nrxns are removed from DA neurons because antibodies that can be used for this in IHC experiments are not available. However, the developmental time course of the DAT gene in mice is very well described and there is no doubt that the deletion occurs during the embryonic period. We added a sentence stating this in the revised discussion (page 22 of the revised manuscript).